# PRKV: PAGE RESTRUCTED KV CACHE FOR HIGH ACCURACY AND EFFICIENCY LLM GENERATION

## ABSTRACT

As the key-value(KV) cache size scales with context length, the substantial GPU memory demand and the overhead of accessing a large KV cache in each decoding step pose challenges for deploying LLMs with long contexts. Among various sparse attention methods, offloading-based KV retrieval method preserves entire KV cache in CPU memory and dynamically retrieves most relevant KV pairs for each decoding step, which performs higher quality and effectively reduces GPU memory footprint than other methods. However, exiting offloading-based KV retrieval methods perform page-level to reduce estimation overhead, which introduces inaccurate KV selection and significant retrieval overhead. We propose PRKV, a framework that both-optimizes algorithm and system for page-level KV retrieval with KV offloading. On the algorithm side, PRKV introduces hybrid KV selection that combines both static and dynamic KV selection strategies. On the system side, PRKV employs contiguous memory indexing and batched transfer optimizations to improve retrieval efficiency. Experiments demonstrate that PRKV improve accuracy across various scenarios and models, delivering up to 6.75× speedup compared to SOTA KV retrieval methods.

## 1 INTRODUCTION

Recent advancements in large language models (LLMs) have significantly enhanced their ability to process long contexts, enabling complex tasks such as multi-document question answering, information retrieval, and code assistance (Wang et al., 2024; Jiang et al., 2024). However, the efficient serving of these long-context LLMs is hampered by two critical challenges related to the key-value (KV) cache, whose size scales with sequence length and inference batch size. Accessing a large KV cache is a memory-bound operation that severely degrades decoding speed. What's more, the substantial GPU memory demand for the KV cache can exceed hardware limits, making it infeasible to store entire long-context sequences.

To mitigate these issues, numerous sparse attention methods have been proposed. One approach, KV eviction, reduces the memory footprint and access latency by discarding tokens deemed unimportant (Xiao et al., 2024b; Zhang et al., 2023). However, its inability to recall evicted tokens leads to irreversible information loss and accuracy degradation in tasks requiring long-range dependency, such as summarization and complex reasoning (Gao et al., 2025; Liu et al., 2025d). Motivated by this drawback, an alternative approach, KV retrieval, has emerged. This method preserves the entire KV cache in CPU memory and dynamically retrieves the most relevant KV pairs for each decoding step. While this method generally achieves higher accuracy than eviction-based methods and effectively reduces GPU memory footprint, it introduces two new concerns.

**Inaccurate KV selection**. KV retrieval requires estimating the importance of each KV pair at every decoding step, token-level approach such as Magicpig and Infinigen (Chen et al., 2025b; Lee et al., 2024)is computationally expensive estimation, as illustrated in Figure 1. To mitigate this overhead, existing methods, such as Quest and ClusterKV (Tang et al., 2024; Liu et al., 2025c), perform retrieval at the page level. However, this coarser granularity results in the selection of less accurate KV pairs compared to the ground-truth top-k selection, as illustrated in Figure 9.

**Significant retrieval overhead**. Limited by the transfer efficiency between CPU memory and GPU memory, retrieval introduces significant latency when fetching sparse KV pairs back to the GPU during decoding.

In this paper, we aim to solve above two concerns and build a inference system for long-context LLM with high accuracy and low latency.

*Reusing top-k KV pairs.* Performing a high-precision top-k KV selection at every decoding step is computationally expensive. However, prior works indicate that the query vectors of consecutively generated tokens are highly similar. This observation motivates the reuse of top-k KV pairs from the previous step. As shown in Figure 3a, combining these reused KV pairs with a local window achieves a high attention recall rate (defined in Section 3.1), resulting in greater selection accuracy with minimal overhead.

*Static and dynamic combined hybrid KV selection.* Although reusing previous top-k KV pairs improves selection accuracy, static selection methods fail to account for the dynamic nature of KV importance (Chen et al., 2024). To address this limitation, we propose a novel hybrid KV selection approach that combines both static and dynamic retrieval strategies during decoding. As illustrated in Figure 3b, our hybrid method achieves a higher recall rate than alternative selection strategies.

*New retrieval pattern with low overhead.* To reduce latency when retrieving KV pairs from CPU memory, we propose a system-level optimization that addresses two key inefficiencies. First, we note that existing head-wise retrieval methods often overlook the critical impact of the KV cache layout on access efficiency. Our analysis shows that the HND layout enables contiguous memory access for head-wise operations, making it ideal for this context. Second, we implement a batched KV transfer mechanism to maximize PCIe bandwidth utilization during retrieval. Together, these optimizations significantly reduce latency.

Based on the preceding analysis, we introduce PRKV, a framework that both-optimizes algorithm and system design for page-level KV retrieval with KV offloading. As illustrated in Figure 4, the algorithmic component of PRKV features a hybrid KV selection method for decoding steps. On the system side, the framework employs contiguous memory indexing and a batched transfer mechanism to maximize retrieval efficiency.

We conducted extensive experiments and ablation studies to evaluate PRKV. In Section 5.1, we assess various LLMs on benchmarks including LongBench V2 (Bai et al., 2024), Needle In A Haystack (Kamradt, 2023), and the reasoning benchmarks AIME (Jia, 2025) and MATH500 (Hendrycks et al., 2021). The results show that PRKV achieves the highest accuracy among other methods and approaches or even exceeds the accuracy of the full cache. In Section 5.2, the end-to-end results demonstrate that PRKV achieves a speedup of up to $6.75\times$ over SOTA KV retrieval methods.

## 2 RELATED WORK

Transformer-based models rely on KV cache for efficient inference. However, as context length increases, KV cache can consume substantial memory, imposing practical limitations on long-sequence processing. To address this, various techniques have been proposed to mitigate the memory consumption of KV cache.

**KV Cache Quantization.** KV Cache Quantization reduce the per-token KV size by approximating high-precision floating points with discrete low-bit values (Liu et al., 2024; Zhang et al., 2024; Hooper et al., 2025), thereby lowering memory footprint of the KV Cache. KIVI (Liu et al., 2024) employs an asymmetric quantization scheme: key caches are quantized channel-wise, while value caches are quantized per token. Quantization methods reduce the KV cache bit width, which are orthogonal to following sparse approach.

**KV Cache Evection.** Eviction-based methods keep a fixed size of KV cache to store the critical KV cache and discard unnecessary ones. Streaming-LLM (Xiao et al., 2024b) retains KV cache only for the initial sink tokens and those within a local window. SnapKV (Li et al., 2024) uses tokens located at the end of the prompts as observation window to identify important tokens, compressing and retaining them with local window tokens. H2O (Zhang et al., 2023) accumulates historical attention scores to estimate token importance and evicts tokens with low historical attention score accordingly. However, evicting tokens prematurely can degrade inference accuracy since tokens initially deemed irrelevant might later become crucial during decoding.

KV Cache retrieval methods retain full KV cache but perform dynamic sparse attention on selected important KV pairs to reduce computation cost for delivering low inference latency. To reduce GPU memory footprint, these approaches typically offload KV cache to the CPU memory.

**Token-Level Retrieval.** These methods estimate important tokens through various techniques and dynamically retrieve important KV cache from CPU. RetrievalAttention (Liu et al., 2025a) and MagicPIG (Chen et al., 2025b) estimate important tokens by approximate nearest neighbor search (ANNS) and Locality Sensitive Hashing (LSH) respectively on CPU. InfiniGen (Lee et al., 2024) additionally computes and stores partial weights and a partial Key cache in GPU, and uses partial Query and partial Key cache to estimate important tokens on GPU during each decoding step. These methods can select important tokens with relative precision. However, the cost of estimating token importance scales linearly with the context length.

**Group-Level Retrieval.** These methods retrieve the KV cache at a group granularity. They compute and store a metadata representation for each group on the GPU, which is used to estimate the group's importance during the decoding step. Prior works such as Quest, Arkvale, and InfLLM (Tang et al., 2024; Chen et al., 2024; Xiao et al., 2024a) partition the sequence into fixed-size pages of contiguous tokens. ClusterKV (Liu et al., 2025c) and RetroInfer (Chen et al., 2025a) employ K-means clustering algorithm to group tokens into semantic clusters and save the cluster centroids as metadata. While these group-level methods improve estimation efficiency, they often select only a few useful tokens.

## 3 OBSERVATION

### 3.1 RECALL RATE DEFINITION

Attention mechanisms are core of transformer-based LLMs. For each attention head $i$, the general attention computation formula is as follows:

$$W_i = \text{softmax}(\frac{Q_i K_i^T}{\sqrt{d}}), \qquad O_i = W_i V_i \tag{1}$$

Here, $W_i$ is the attention scores across tokens, and $O_i$ is the attention output from the $i$-th attention head. Sparse attention algorithms aim to retain the most important tokens and achieve the highest attention scores with a fixed budget $k$ through different estimation methods, which can be presented as:

$$I_i = \text{argtopk}\left(f(Q_i, K_i),\ k\right) \tag{2}$$

where $I_i$ denotes the selected subset tokens' indices in $i$-th attention head, and the function $f$ is an efficient estimation method to identify the importance of the tokens. The **Attention Recall Rate** is defined as the proportion of the ground truth attention scores that the selected subset covers:

$$\mathcal{R}_i = \sum_{j \in I_i} W_{ij} \tag{3}$$

For a given KV budget $k$, ground truth top-k means select $k$ KV pairs with highest ground truth attention scores, which guarantees the highest recall rate in return. A high attention recall rate demonstrates that the selected tokens encompass the most significant attention scores, which is essential to maintain high accuracy.

### 3.2 LONG ESTIMATION TIME OF TOKEN-LEVEL METHODS

In long-context scenarios, KV cache is often offloaded to CPU memory due to GPU memory constraints. Prior token-level retrieval methods, such as MagicPIG (Chen et al., 2025b), estimate token importance by computing $W_i$ using the full Key cache. This requires computing the entire Key cache in CPU, incurring an $O(L)$ computational complexity (where $L$ is the context length) and significant CPU processing overhead. In contrast, group-level methods like Arkvale (Chen et al., 2024) operate more efficiently. They leverage page-level metadata (e.g., page digest) stored in GPU

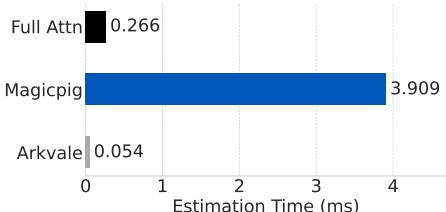

Figure 1: Estimation time of MagicPIG and Arkvale with 64K sequence length

memory to estimate token importance at a page granularity. To compare their efficiency, we measured the estimation time of MagicPIG and Arkvale with a 64K sequence length. As shown in Figure 1, the estimation time of MagicPIG is $15\times$ higher than full cache attention with FlashAttention2 (Dao, 2023), whereas Arkvale's estimation time is significantly lower.

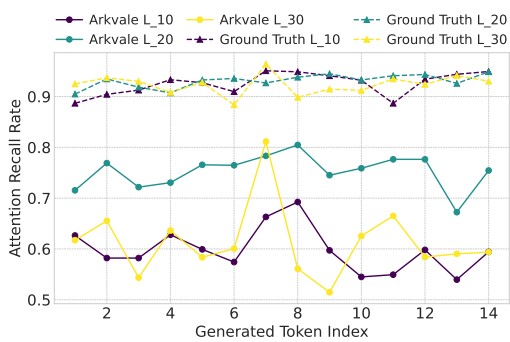 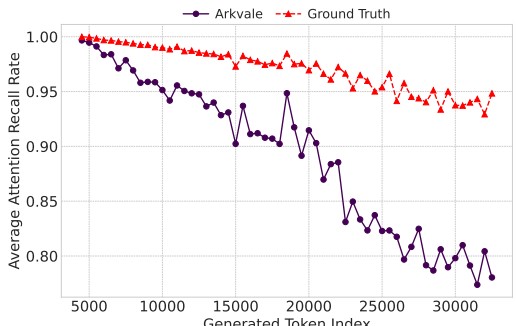

(a) Attention recall rate of different approaches using 1K token budget on a 37K NIAH task

(b) Attention recall rate of different approaches using 4K token budget on an AIME TASK

Figure 2: Analysis of Attention Recall Rate between Arkvale and ground truth topk on different tasks. Figure 2a shows attention recall rate of layer10, layer20 and layer30 on an NIAH(Kamradt, 2023) task, using a token budget of 1K and input length is 37K on Llama3.1-8b-Instruct. Figure 2b shows average attention recall rate across all layers on an AIME(Jia, 2025) problem, using a token budget of 4K and generation length up to 32K tokens on Qwen3-8B.

### 3.3 LIMITATION OF PAGE-LEVEL METHODS

Existing page-level selection methods reduce estimation cost by grouping consecutive tokens into pages, and it is also highly compatible with modern KV cache management frameworks like PageAttention and FlashInfer (Kwon et al., 2023; Ye et al., 2025). However, current page-level selection methods overlook the fact that tokens within the same page can exhibit vastly different importance though they have high semantic similarities. As shown in Figure 9, important tokens selected by ground truth topk are distributed throughout the sequence, while Arkvale's page grained nature fails to select these important individual tokens, directly resulting in a lower accuracy.

We compare the **Attention Recall Rate** of the ground-truth top-k with that of Arkvale, a SOTA page-level selection method, under both long input and long generation scenarios, respectively. As shown in Figure 2a, across all layers in the NIAH task, Arkvale's attention recall rate shows a significant shortfall, averaging 25% below the ground-truth performance. Similarly, on the AIME2024 task, it achieves a recall rate of only approximately 80% while ground-truth has an average of 95% recall rate (Figure 2b). It is crucial to note that this Attention Recall Rate is a theoretical metric calculated against the full KV cache. Consequently, in a practical sparse inference setting, the bias introduced by low attention recall rate at each decoding step can accumulate throughout the generation process, ultimately leading to a degradation in inference quality.

### 3.4 OPPORTUNITY1: REUSE TOP-k TOKENS WITH LOCAL TOKENS

Prior studies have shown that the query vectors of consecutive generated tokens exhibit high similarity, likely due to position embeddings (Su et al., 2024) and the semantic continuity of adjacent tokens (Yuan et al., 2025). Leveraging this property, some works like FreeKV (Liu et al., 2025b) reuse top-k tokens' KV pairs from previous step. As shown in Figure 3a, we set the KV budget (i.e., top-k) to 1024. While reusing only top-k tokens from relative position 0 yields an attention recall rate of 0.65, and the rate rises to a level nearly matching the ground truth top-k when additional 64 local tokens are added (i.e., reuse w/ local tokens). This observation suggests that reusing top-k tokens' KV together with local window can achieve a higher attention recall rate.

### 3.5 OPPORTUNITY2: HYBRID SPARSE TOKEN SELECTION

As described above, reusing the top-k KV pairs of previous step offers a fine-grained method for token-level selection. We investigate the potential of integrating such a fine-grained method into dynamic page-level selection framework, without token eviction. Specifically, for a given KV budget $k$, we partition it into two complementary parts. The first is a static set of tokens, comprising a

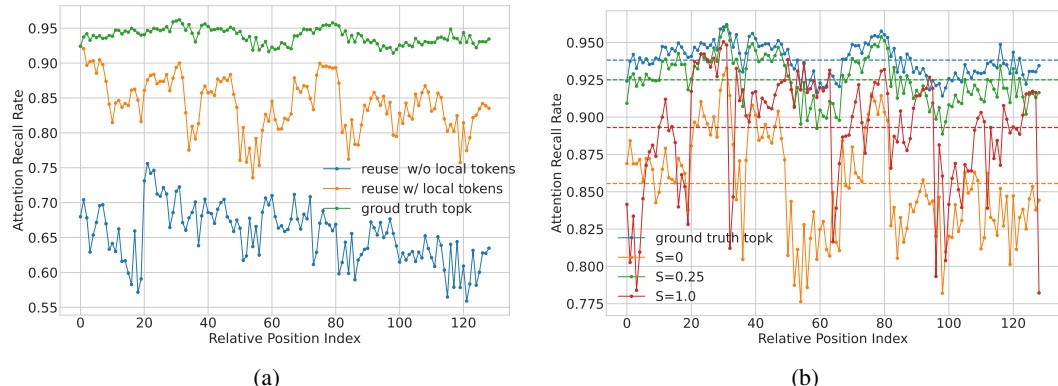

(a)                                                     (b)

Figure 3: Average Attention Recall Rate conducted on sampled LongBench tasks with 32K input length. (a) 'reuse w/o local KVs' represents only reusing top-k (k=1024) KV pairs started from relative position 0; 'reuse w/ local KVs' represents additionally using local 64 KV pairs at the end of the sequence. (b) Attention Recall Rate of different $\mathcal{S}$ settings with 1K budget

proportion of the budget (the ratio is denoted as $\mathcal{S}$), which consists of the top $k \times \mathcal{S}$ KV pairs from previous step. This part participates in sparse attention with local tokens during subsequent decoding step. The second part is a dynamic set of pages selected from the remaining KV pairs based on their importance, which are involved in the subsequent decoding step at page granularity. The ratio of KV pairs in the second part account for $1 - \mathcal{S}$. We compare the Attention Recall Rates of such a hybrid setting on LongBench tasks, with the results presented in Figure 3b. In this experiment, we start the selection methods at relative position 0. As the comparison, both fully dynamic page-level selection and fully static selection are evaluated as well. $\mathcal{S} = 0$ represents a fully dynamic page-level selection (e.g., Arkvale), while $\mathcal{S} = 1$ corresponds to a fully static selection (e.g., SnapKV). For our hybrid selection method, we take $\mathcal{S} = 0.25$ as an example. The results demonstrate that a hybrid selection method achieves a higher recall rate than both the fully static selection method and the fully dynamic selection method. This experimental observation supports the design of **hybrid selection** that combines static and dynamic KV pair retrieval in decoding sparse attention.

## 4  PRKV

Based on the above analysis and observations, we propose Page Restructed KV Cache (PRKV), a method combines static and dynamic KV pair retrieval in page granularity, achieving higher Attention Recall Rate and lower retrieval overhead.

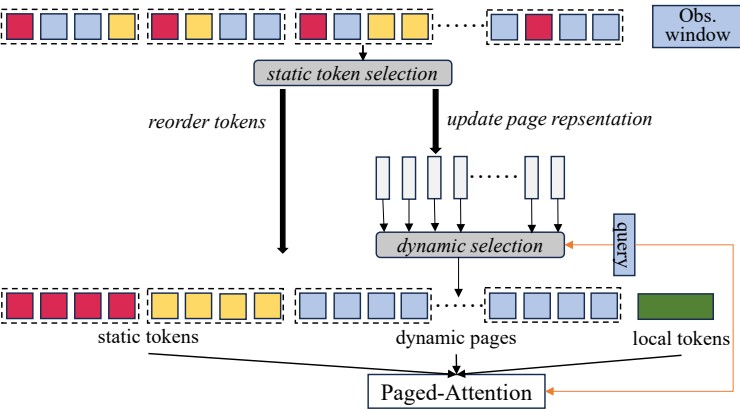

Figure 4: Algorithm design overview of PRKV

## 4.1 OBSERVATION BASED HYBRID TOKEN SELECTION

**Static token-level selection.** The detailed mechanism of static token-level selection algorithm is formalized in Algorithm 1 lines 6-14. As discussed in Section 3.4, the local tokens are consistently important in sparse attention computation, so PRKV keeps the local tokens at the end of context. Similar to SnapKV, PRKV uses an observation window at the end of sequence(i.e., the local window). It then computes the attention scores between the queries in the observation window and the keys of all preceding tokens. Finally, PRKV selects the top-k tokens with the highest attention scores as the static tokens. After identifying static tokens, PRKV relocates their corresponding KV to the beginning of KV cache to make them contiguous, while preserving the relative order of the remaining tokens and still organizing KV cache to pages. PRKV recomputes and updates page representations since the KV cache pages have been restructured(line 9-14).

To maintain compatibility with Grouped-Query Attention (GQA) and achieve more precise token selection, PRKV modifies the process of computing observation-based attention scores. Averaging queries within a group before computation can be problematic, as positive and negative scores from different query heads may average to near zero and mask token importance. Instead, PRKV first repeats the group's Key cache for each query head. It then computes the attention score for each head individually and, finally, averages these scores within the group.

**Periodic static token update.** The importance of initially selected static tokens can decay over long decoding steps, as shown in Figure 3, the recall rate of reusing tokens steadily decreases as the generation length increases. To mitigate this, we introduce a periodic update strategy for the static token set, formalized in Algorithm 1, lines 4-7.

**Hybrid KV Sparsity.** PRKV offloads all KV cache to CPU memory, keeping page representations in GPU memory for page importance estimation. Similar to Arkvale, these representations are bounding volumes that form a convex set around the keys, allowing the model to approximate the maximum token importance in a page. During each decoding step, for a KV budget $k$, PRKV selects a combination of static tokens, local tokens, and a set of $\frac{kS}{pagesize}$ dynamically chosen pages. These selected tokens' KV then participate in the sparse paged-attention computation (Kwon et al., 2023).

---

**Algorithm 1** Static Token Selection

1: **Input:** querys of observation window$Q\_obs$, key value cache $K$ and $V$, token budget $k$, static ratio $S$, decode step $ds$, update interval $T$
2: **Constants:** query group size $group\_size$, local window size $n\_local$
3: **Output:**
4: **if** $ds\%T \neq 0$ **then**
5:     **return** $K, V, None$
6: $K'_{pre} \leftarrow$ repeat_kv($K[: -n\_local], group\_size$)       ▷ Repeat KV heads to match GQA
7: $S \leftarrow$ compute_score($Q_{obs}, K'_{pre}$)       ▷ Observation window attention scores
8: $S_{mean} \leftarrow$ Average($S$)     ▷ Compute average scores in group and observation window dim
9: $I_{\text{topk}} \leftarrow$ TopKIndices($S_{mean}, kS$)       ▷ select static important tokens
10: $I_{\text{rem}} \leftarrow$ AllIndices $\setminus I_{\text{topk}}$
11: $P \leftarrow$ Concat($I_{\text{topk}}, I_{\text{rem}}$)       ▷ Put static set indices in front of sequence
12: $K[: -n\_local] \leftarrow$ gather($K[: -n\_local], P$)
13: $V[: -n\_local] \leftarrow$ gather($V[: -n\_local], P$)
14: $Rep_{page} \leftarrow$ update_page_representation($K[:-n\_local]$)
15: **return** $K, V, Rep_{page}$

---

## 4.2 EFFICIENT RETRIEVAL OPTIMIZATION

Identifying critical key-value (KV) pairs adds latency to the inference process, as it resides on the critical path. This issue is exacerbated with long-sequence scenarios, where the KV cache is often offloaded to CPU memory. Retrieval efficiency in this case is bottlenecked by the PCIe bus. Even when fetching only a small set of important (or reconstructed) pages, significant costs are incurred from two sources: inefficient KV indexing and data transfer. The indexing overhead arises because the KV layout in CPU memory may not align with the physical memory block ("block" denoting

the physical memory page) layout. Consequently, reconstructing a single KV page can require a time-consuming search across numerous physical blocks. The transfer overhead occurs because moving multiple non-contiguous KV pages necessitates launching multiple GPU kernels and serial transfers, which is highly inefficient. We therefore implement an efficient KV cache reconstructing and transfer mechanisms by designing a custom Triton kernel.

**Contiguous Memory Indexing.** When managing KV pages in CPU memory, two common layouts are NHD and HND, with tensor shapes of $(2, page\_size, num\_head, head\_dim)$ and $(num\_head, 2, page\_size, head\_dim)$, respectively. These layouts differ significantly in indexing efficiency for page retrieval, which is performed per head. Retrieving a page for head $i$ under the NHD layout requires indexing all KV pairs within a page to copy those for head $i$. In contrast, the HND layout allows direct indexing of all KV pairs for a specific head without search overhead. Consequently, PRKV adopts the HND layout for both CPU and GPU KV caches, utilizing contiguous memory indexing. The efficiency of this design is demonstrated in Section 5.3.

**Batched KV Transfer.** As shown on the left of Figure. 22, the naive page-to-page transfer mode moves KV data directly from CPU memory to GPU memory. This approach, however, requires multiple kernel launches, which is inefficient. To address this, we introduce Batched KV Transfer (right side of Figure. 22). Our method first consolidates the required KV pages into a pinned CPU buffer, then performs a single, bulk transfer to GPU memory. Finally, the data is dispatched into the GPU's KV pool.

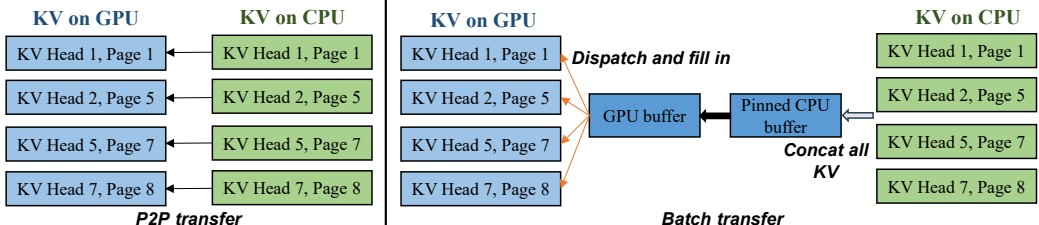

Figure 5: KV transfer optimization

# 5 EXPERIMENTS

## 5.1 ACCURACY EVALUATION

**Setup.** We evaluate PRKV across various models and tasks. We conduct evaluation on LongBench V2 (Bai et al., 2024), Needle In A Haystack (Kamradt, 2023), and long reasoning tasks, including MATH500 (Hendrycks et al., 2021) and AIME24 (Jia, 2025). For LongBench V2 and NIAH, we use general models including Llama-3.1-8B-Instruct(Meta Llama, 2024) and Qwen-2.5-14B-Instruct (Qwen et al., 2025). For reasoning tasks, we use Qwen3-8B and Qwen3-4B (Yang et al., 2025).

**Baselines** We compare PRKV against four dynamic sparse attention methods, including Quest (Tang et al., 2024), Arkvale, ShadowKV (Sun et al., 2024) and SnapKV. All methods perform dynamic sparse attention during decoding. For Quest, Arkvale and PRKV, we set their page size to 32 and local window size to 64. For PRKV the staic set ratio $\mathcal{S} = 0.25$ and update interval $T = 128$.

**LongBench V2.** LongBench V2 covers various difficulty levels and context lengths, ranging from 8K to 2M tokens. For all methods, if the input length exceeds the model's maximum length, we truncate the input by taking half of the max_len from the beginning and half from the end, and concatenate them to form the final input. The average scores is shown in Figure 6a and Figure 6b. PRKV consistently achieves the highest score. While other methods experience serious performance degradation under a low KV budget, PRKV is more robust and even outperforms original full attention when tested on Qwen-2.5-14B-Instruct. The detail results of all kind of tasks are shown in Appendix E.1.

**Needle In A Haystack.** In the NIAH benchmark, a single critical sentence (the "needle") is placed within a large, mostly irrelevant context. The task requires the model to retrieve this key sentence and answer a related question. We evaluate the method using the retrieval accuracy metric. We use KIMI

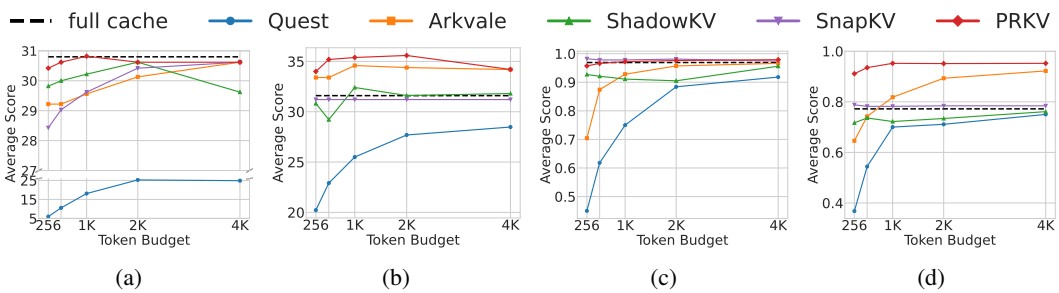

Figure 6: Average Score of PRKV and other baselines. (a) LongBench V2 on Llama3.1-8B-Instruct. (b) LongBench V2 on Qwen-2.5-14B-Instruct. (c) NIAH on Llama3.1-8B-Instruct. (d) NIAH on Qwen-2.5-14B-Instruct.

API to generate the retrieval accuracy by providing it with the question, the retrieval answer, and the test model's response. The average scores is shown in Figure 6c and Figure 6d. On Llama3.1-8B-Instruct, PRKV and SnapKV achieve highest scores, closely matching the performance of the full cache. Meanwhile, on Qwen-2.5-14B-Instruct, PRKV constantly out perform than full cache. The detail heatmaps are provided in Appendix E.2.

**Reasoning tasks.** We evaluate reasoning tasks using problems from MATH500, AIME24 datasets. In this test we consider two widely-used reasoning models, Qwen3-8B and Qwen3-4B. we set the maximum generation length to 32K, and apply sampling temperature with 0.6 and a top-p value of 0.95. Following existing works (DeepSeek-AI et al., 2025), we utilize pass@k evaluation (Chen et al., 2021) and report two metrics: pass@k, which measures the likelihood of at least one correct solution among the k samples, avg@k, which represents the average accuracy across all k samples. Here we set k = 4, which means we generate 4 different samples for each problem. We set KV budget to 1024 for all the KV retrieval methods, since SnapKV originally designed for long prefilling and to adapt it for decoding, we apply the same static tokens update interval as our method.

Table 1: Accuracy results of long reasoning tasks.

| Methods | AIME24 | | MATH500 | |
|---|---|---|---|---|
| | *pass@k* | *avg@k* | *pass@k* | *avg@k* |
| *Qwen3-4B* | 86.67 | 74.17 | 95.40 | 94.20 |
| Quest | 70.00 | 56.67 | 84.80 | 78.50 |
| ArkVale | 76.67 | 62.50 | 85.40 | 84.30 |
| ShadowKV | 76.67 | 59.17 | 85.00 | 81.10 |
| SnapKV | 83.33 | 67.50 | 93.60 | 90.00 |
| PRKV | **86.67** | **75.00** | **95.00** | **93.80** |
| *Qwen3-8B* | 83.33 | 74.17 | 96.20 | 95.10 |
| Quest | 76.67 | 65.83 | 86.80 | 78.60 |
| ArkVale | 80.00 | 68.33 | 87.00 | 86.90 |
| ShadowKV | 76.67 | 67.50 | 86.80 | 82.60 |
| SnapKV | **83.33** | 67.50 | 93.60 | 93.60 |
| PRKV | **83.33** | 73.33 | 95.40 | 94.40 |

As shown in Table 1, PRKV delivers accuracy comparable to models with full KV cache and outperforms other compression methods across all datasets. Moreover, SnapKV achieves higher accuracy than other page-level retrieval methods, which demonstrates that its mechanism of reusing important tokens be highly effective, provided that the static token set is periodically updated. Furthermore, PRKV consistently outperforms other methods, demonstrating the high accuracy of its hybrid KV selection.

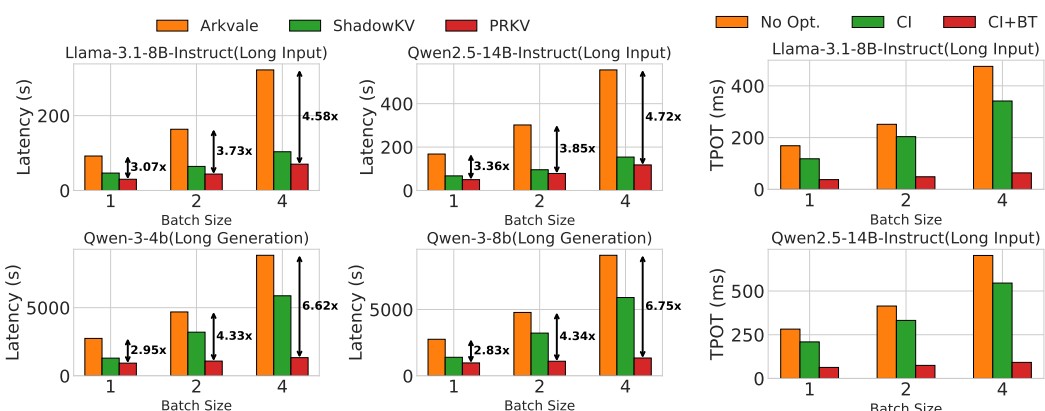

Figure 7: **Left:** End-to-end latency under long-input and long-generation scenarios. **Right:** TPOT of different system optimization design.

## 5.2 EFFICIENCY EVALUATION

The efficiency experiments were conducted on an single compute unit connected with Intel CPU via PCIe Gen4. We compare PRKV with offloading methods Arkvale and ShadowKV under both long-input (32K input, 512 output) and long-generation scenarios (400 input, 16K output). We set KV budget $k = 1024$. The end-to-end latency is in the left of Figure. 7. PRKV achieves a multi-fold speedup over Arkvale and ShadowKV across multiple models, demonstrating significant efficiency and achieving maximum speedup of up to up to $6.75\times$ compared to ArkVale.

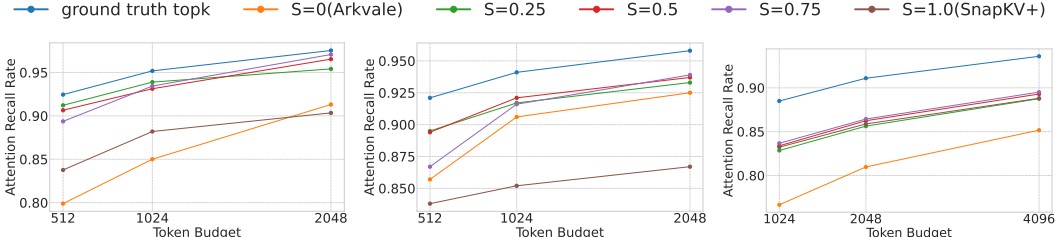

Figure 8: **Left:** Average Attention Recall Rate on AIME24 task with 16K generation. **Middle:** Average Attention Recall Rate on Longbench V2 task with 64K input and 512 generation. **Right:** Average Attention Recall Rate on NIAH task with 128K input.

## 5.3 ABLATION RESULTS

Static Ratio, local window size, update T **Static set ratio.** We compare the Attention Recall Rate for different static set ratios on sampled tasks from LongBench V2 and AIME24 across various across various KV budget. We test five settings for the static set ratio, $\mathcal{S}$ : $0, 0.25, 0.5, 0.75, 1.0$. Here $\mathcal{S} = 0$ represents a fully dynamic selection (equivalent to *Arkvale*), while $\mathcal{S} = 1.0$ represents a fully static reuse strategy (equivalent to SnapKV with the full cache reserved and updated periodically, named *SnapKV+*). The average results is shown in Figure 8. It is obvious that hybrid KV selection with different $\mathcal{S}$ all perform high Attention Recall Rate than fully dynamic selection or fully static reuse strategies. In the AIME task, the performance of $\mathcal{S} = 0.25$ The detailed results are shown in Section F.

**System optimization.** Since PRKV's system optimizations including contiguous memory indexing (CI) and batched KV transfer (BT) are designed to reduce recall latency, we evaluate the Time Per Output Token (TPOT) using Llama3.1-8B-Instruct and Qwen2.5-14B-Instruct with 512 generated tokens. As shown right of Figure. 7, contiguous memory indexing achieve $1.5\times$ reduction in TPOT. Batched KV transfer contributes most to the improvements, achieving up to a $7.7\times$ TPOT reduction.

## 6 ETHICS STATEMENT

We believe this work raises no ethical concerns. Attention is a key component in Transformers, widely used in Large Language Models (LLMs). Therefore, accelerating the execution of attention of decoding, especially in long context, is beneficial for developing LLM applications that address diverse societal challenges.

## 7 REPRODUCIBILITY STATEMENT

All of the experiments are reproducible in supplementary materials, including observation test.

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

## A THE USE OF LARGE LANGUAGE MODELS(LLMS)

We use Large Language Models (LLMs) in polish writing and do some translation works.

## B ETHICS STATEMENT

We believe this work raises no ethical concerns. Attention is a key component in Transformers, widely used in Large Language Models (LLMs). Therefore, accelerating the execution of attention, especially in long context, is beneficial for developing LLM applications that address diverse societal challenges.

## C REPRODUCIBILITY STATEMENT

All of the experiments are reproducible in our anonymous repository. All of the code of ours and other methods can be found in the supplementary materials.

## D PICTURE OF OBSERVATION

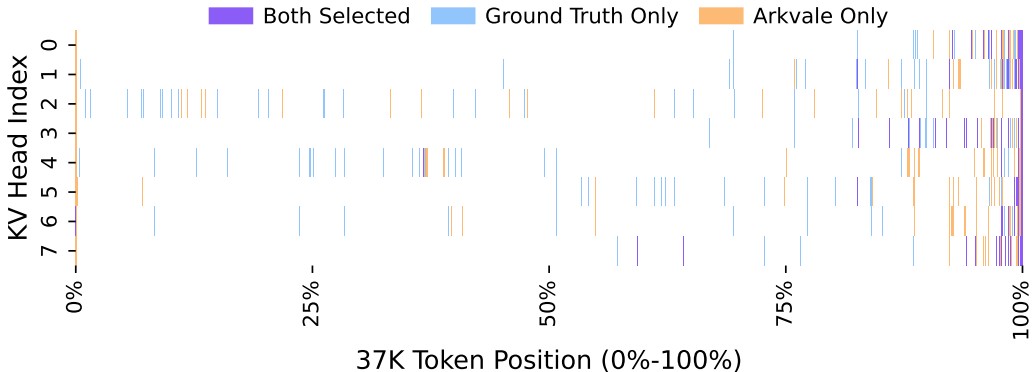

Figure 9: Important token selection distribution of ground truth top-k and Arkvale on a NIAH task.

Figure 9 shows the important token selection distribution of ground truth topk and Arkvale on a 37K NIAH task, using a 1K token KV budget. The result is from layer 5 of the 14-th generation token.

## E DETAIL RESULTS IN EVALUTION

In this section, we provide detail results in evalution.

### E.1 LONGBENCH V2

We report accuracy under the context length categories of short, medium and long, which has easy and hard two level problems, as well as the overall accuracy. As shown in Table 2 and Table 6

### E.2 NIAH

We show the detail results of NIAH across Llama-3.1-8B-Instruct and Qwen-2.5-14B-Instruct. We test the KV budget from 256, 512, 1024, 2048, 4096.

Table 2: Accuracy results of LongBench v2

|  |  | Overall | Easy | Hard | Short | Medium | Long |
|---|---|---|---|---|---|---|---|
| *Llama-3.1-8B-Instruct (base)* | | 30.8 | 30.7 | 30.9 | 38.3 | 27.0 | 25.9 |
| *k=4096* | Quest | 24.85 | 22.92 | 26.05 | 30.0 | 21.86 | 22.22 |
| | ArkVale | 30.62 | 28.65 | 31.83 | 36.11 | 27.91 | 26.85 |
| | ShadowKV | 29.62 | 29.17 | 29.9 | 33.89 | 28.84 | 24.07 |
| | SnapKV | 30.42 | 28.65 | 31.51 | 33.89 | 30.23 | 25.0 |
| | PRKV | **30.62** | 28.12 | 32.15 | 36.67 | 28.84 | 24.07 |
| *k=2048* | Quest | 25.25 | 22.92 | 26.69 | 31.11 | 23.72 | 18.52 |
| | ArkVale | 30.13 | 31.49 | 29.27 | 36.05 | 27.04 | 26.0 |
| | ShadowKV | 30.62 | 29.69 | 31.19 | 36.11 | 28.37 | 25.93 |
| | SnapKV | 30.42 | 28.12 | 31.83 | 36.67 | 28.37 | 24.07 |
| | PRKV | **30.62** | 28.65 | 31.83 | 37.22 | 28.84 | 23.15 |
| *k=1024* | Quest | 18.09 | 19.27 | 17.36 | 18.89 | 17.67 | 17.59 |
| | ArkVale | 29.56 | 31.29 | 28.51 | 37.12 | 25.43 | 26.19 |
| | ShadowKV | 30.22 | 27.6 | 31.83 | 33.9 | 29.3 | 25.9 |
| | SnapKV | 29.62 | 28.12 | 30.55 | 35.56 | 27.44 | 24.07 |
| | PRKV | **30.82** | 30.21 | 31.19 | 35.56 | 29.77 | 25.0 |
| *k=512* | Quest | 10.54 | 9.9 | 10.93 | 12.22 | 9.77 | 9.26 |
| | ArkVale | 29.22 | 28.65 | 29.58 | 33.89 | 27.91 | 24.07 |
| | ShadowKV | 30.0 | 27.6 | 31.5 | 34.4 | 28.4 | 25.9 |
| | SnapKV | 29.03 | 28.12 | 29.58 | 36.67 | 26.05 | 22.22 |
| | PRKV | **30.62** | 31.25 | 30.23 | 36.11 | 29.77 | 23.15 |
| *k=256* | Quest | 5.96 | 3.65 | 7.4 | 8.33 | 5.58 | 2.78 |
| | ArkVale | 29.22 | 27.6 | 30.23 | 33.89 | 27.91 | 24.07 |
| | ShadowKV | 29.82 | 29.17 | 30.23 | 36.67 | 28.37 | 21.3 |
| | SnapKV | 28.43 | 25.52 | 30.23 | 34.44 | 25.58 | 24.07 |
| | PRKV | **30.42** | 28.65 | 31.51 | 33.89 | 30.23 | 25.0 |

Table 3: Accuracy results of LongBench v2

|  |  | Overall | Easy | Hard | Short | Medium | Long |
|---|---|---|---|---|---|---|---|
| *Qwen-2.5-14B-Instruct (base)* | | 31.6 | 35.4 | 29.3 | 37.2 | 30.2 | 25.0 |
| $k=4096$ | Quest | 28.48 | 30.9 | 26.95 | 33.73 | 25.25 | 26.04 |
| | ArkVale | 34.19 | 41.15 | 29.9 | 37.78 | 34.88 | 26.85 |
| | ShadowKV | 31.81 | 33.85 | 30.55 | 41.67 | 27.44 | 24.07 |
| | SnapKV | 31.21 | 33.33 | 29.9 | 38.33 | 29.3 | 23.15 |
| | PRKV | **34.2** | 40.1 | 30.5 | 38.9 | 34.4 | 25.9 |
| $k=2048$ | Quest | 27.69 | 28.27 | 27.33 | 35.75 | 23.26 | 23.15 |
| | ArkVale | 34.39 | 41.67 | 29.9 | 38.89 | 34.42 | 26.85 |
| | ShadowKV | 31.61 | 34.9 | 29.58 | 39.44 | 28.37 | 25.0 |
| | SnapKV | 31.21 | 33.33 | 29.9 | 38.33 | 29.3 | 23.15 |
| | PRKV | **35.59** | 42.71 | 31.19 | 38.89 | 35.81 | 29.63 |
| $k=1024$ | Quest | 25.5 | 25.65 | 25.4 | 29.61 | 26.98 | 15.74 |
| | ArkVale | 34.59 | 40.62 | 30.87 | 39.44 | 34.42 | 26.85 |
| | ShadowKV | 32.41 | 35.94 | 30.23 | 39.44 | 29.77 | 25.93 |
| | SnapKV | 31.21 | 33.33 | 29.9 | 38.33 | 29.3 | 23.15 |
| | PRKV | **35.39** | 43.75 | 30.23 | 38.33 | 35.81 | 29.63 |
| $k=512$ | Quest | 22.91 | 26.18 | 20.9 | 26.82 | 23.26 | 15.74 |
| | ArkVale | 33.4 | 38.02 | 30.55 | 37.22 | 33.49 | 26.85 |
| | ShadowKV | 29.22 | 31.77 | 27.65 | 36.11 | 27.91 | 20.37 |
| | SnapKV | 31.21 | 33.33 | 29.9 | 38.33 | 29.3 | 23.15 |
| | PRKV | **35.19** | 41.67 | 31.19 | 39.44 | 34.42 | 29.63 |
| $k=256$ | Quest | 20.22 | 24.72 | 17.38 | 22.89 | 20.2 | 15.62 |
| | ArkVale | 33.4 | 40.62 | 28.94 | 38.33 | 33.02 | 25.93 |
| | ShadowKV | 30.82 | 33.33 | 29.26 | 38.89 | 28.37 | 22.22 |
| | SnapKV | 31.21 | 33.33 | 29.9 | 38.33 | 29.3 | 23.15 |
| | PRKV | **34.0** | 42.19 | 28.94 | 36.11 | 34.42 | 29.63 |

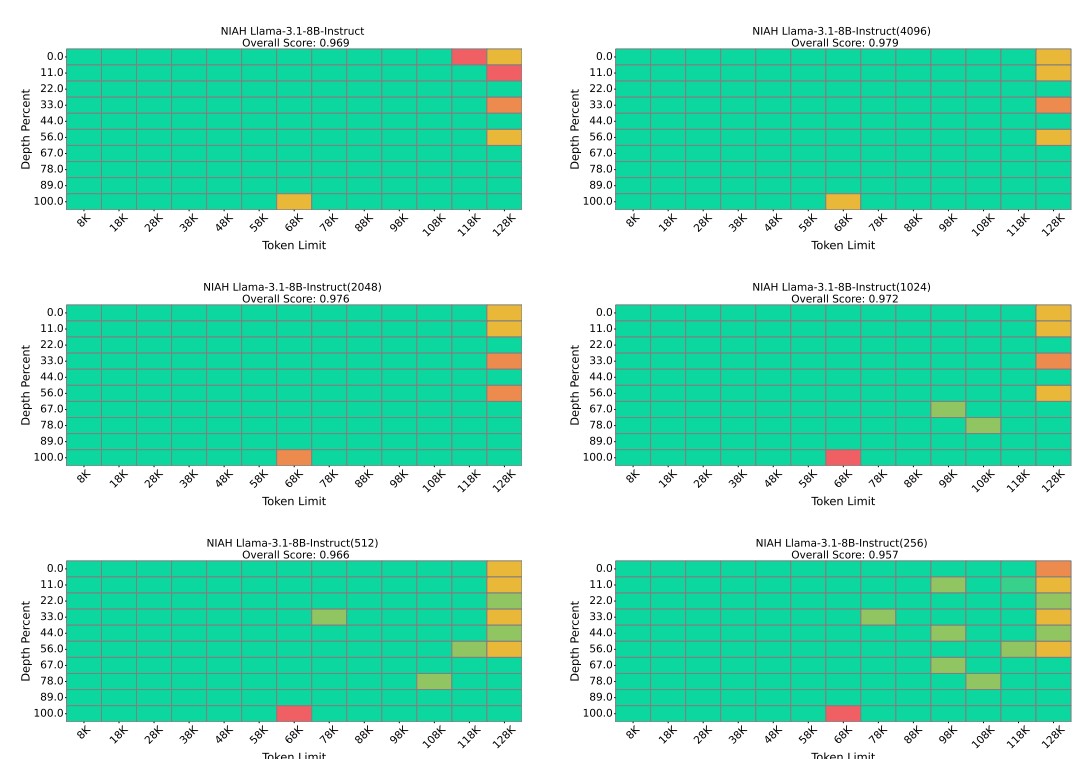

Figure 10: Compare all KV budget in PRKV with full cache in Llama-3.1-8B-Instruct

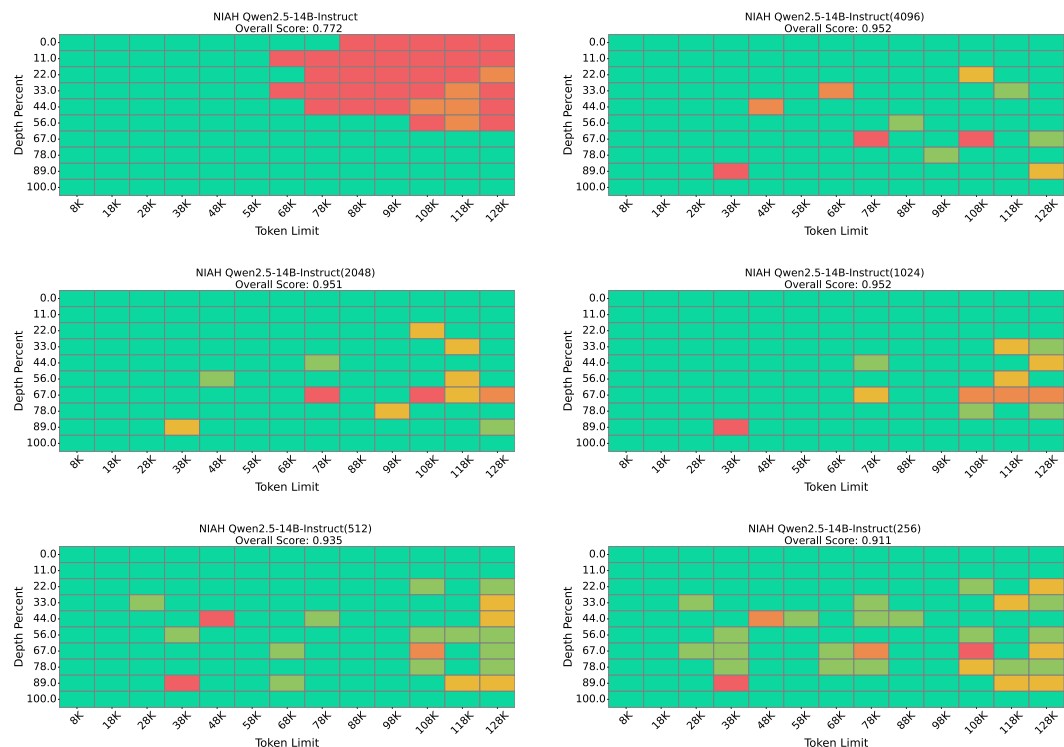

Figure 11: Compare all KV budget in PRKV with full cache in Qwen2.5-14B-Instruct

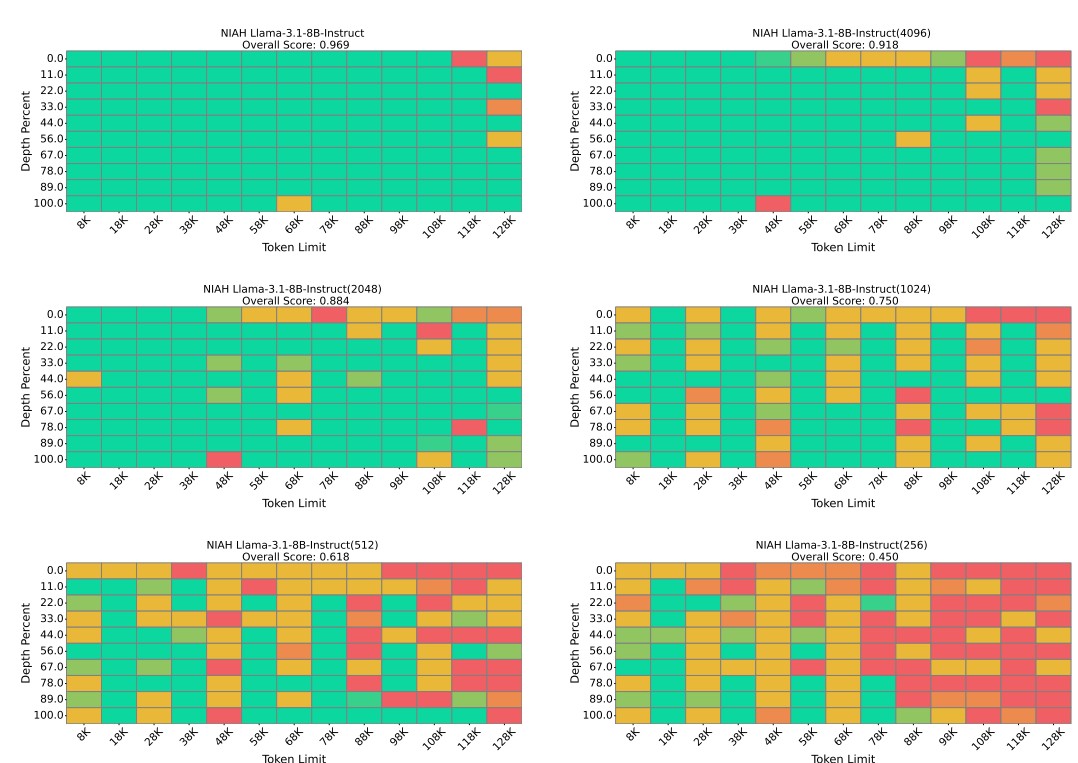

Figure 12: Compare all KV budget in Quest with full cache in Llama-3.1-8B-Instruct

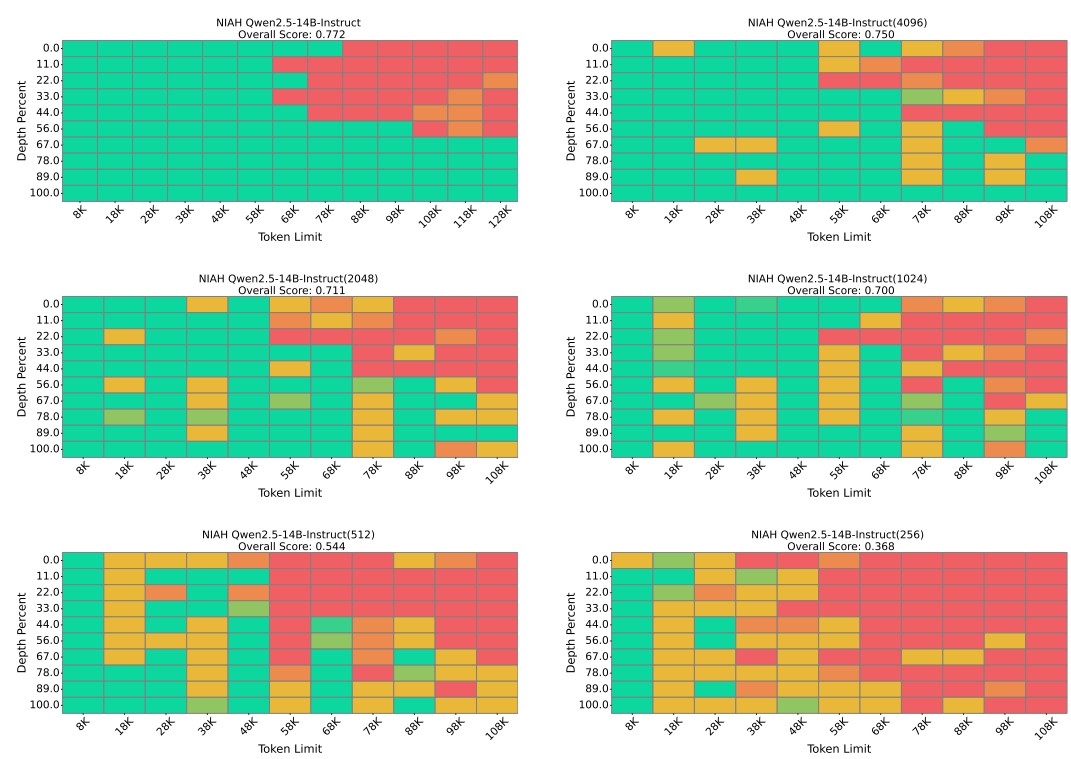

Figure 13: Compare all KV budget in Quest with full cache in Qwen2.5-14B-Instruct

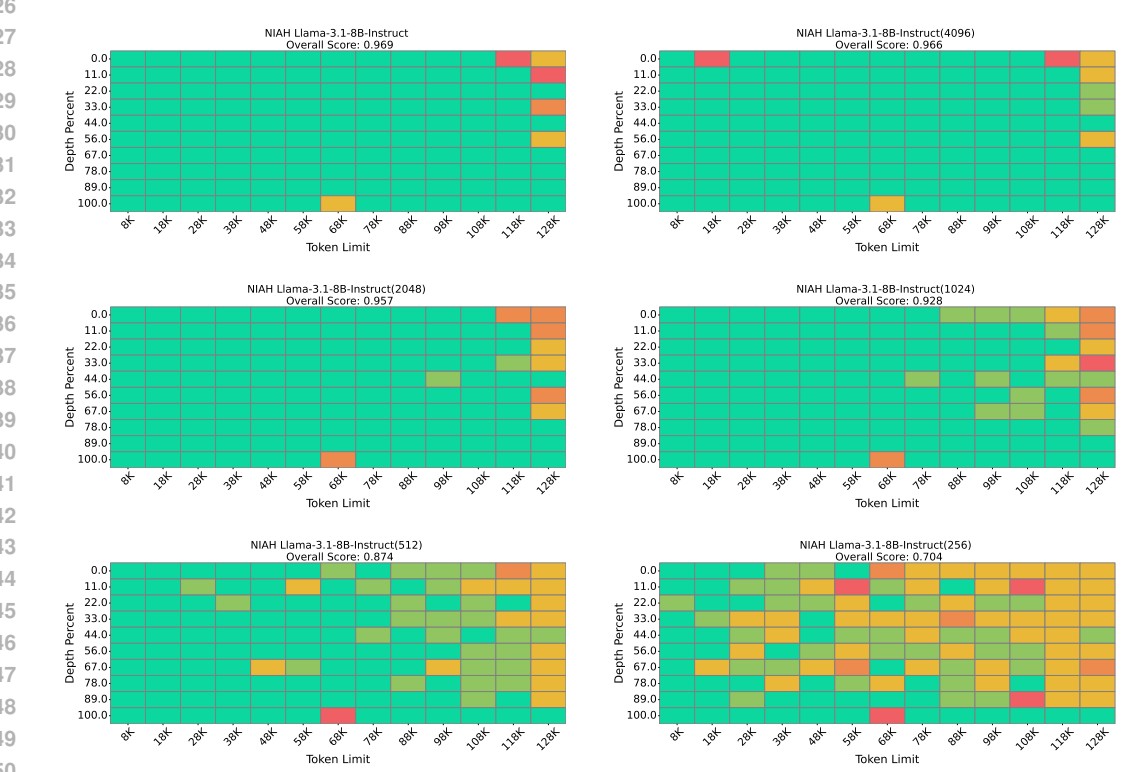

Figure 14: Compare all KV budget in Arkvale with full cache in Llama-3.1-8B-Instruct

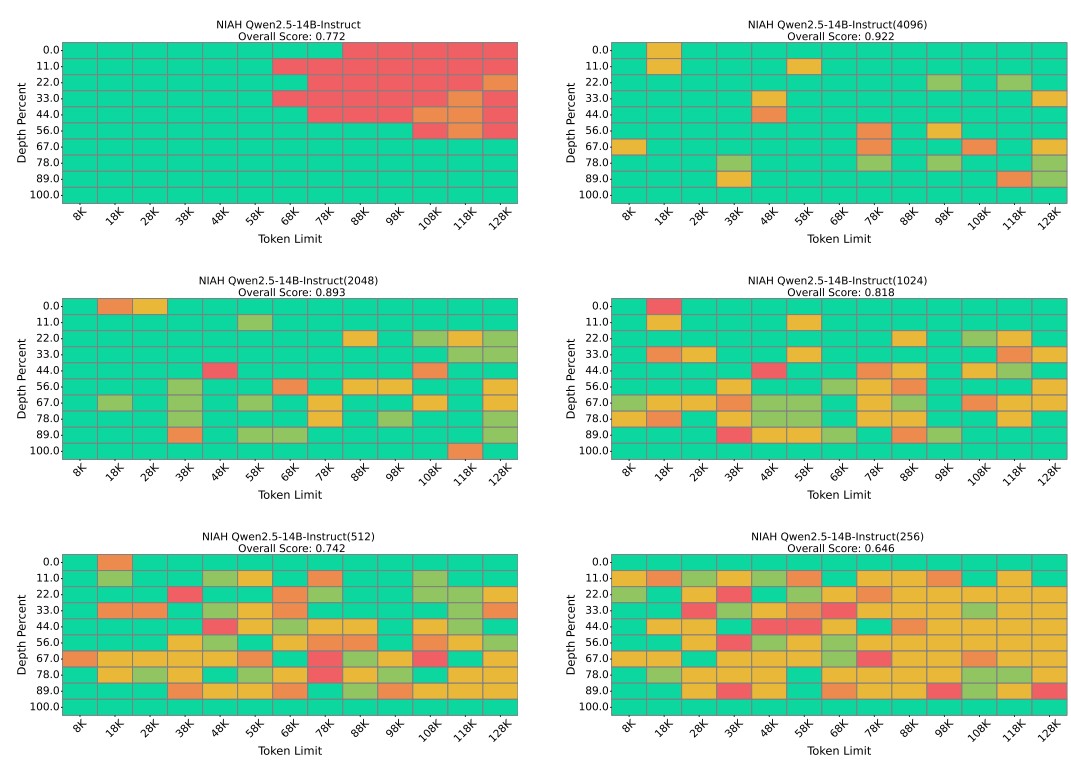

Figure 15: Compare all KV budget in Arkvale with full cache in Qwen2.5-14B-Instruct

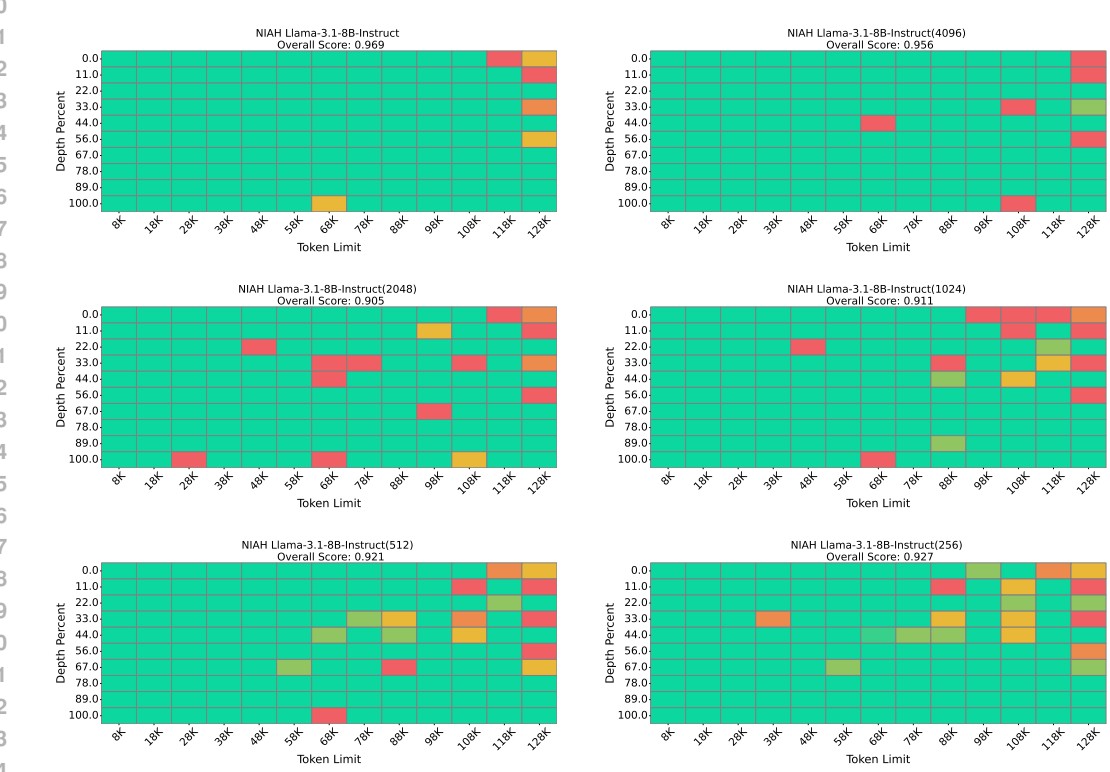

Figure 16: Compare all KV budget in ShadowKV with full cache in Llama-3.1-8B-Instruct

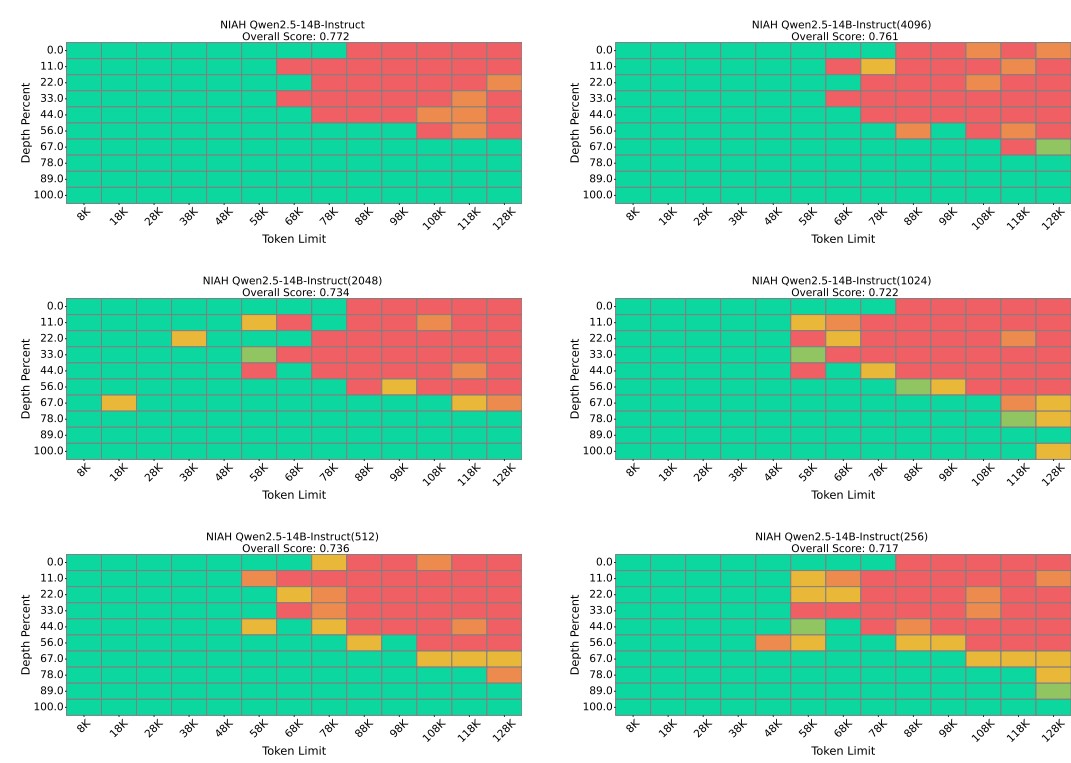

Figure 17: Compare all KV budget in ShadowKV with full cache in Qwen2.5-14B-Instruct

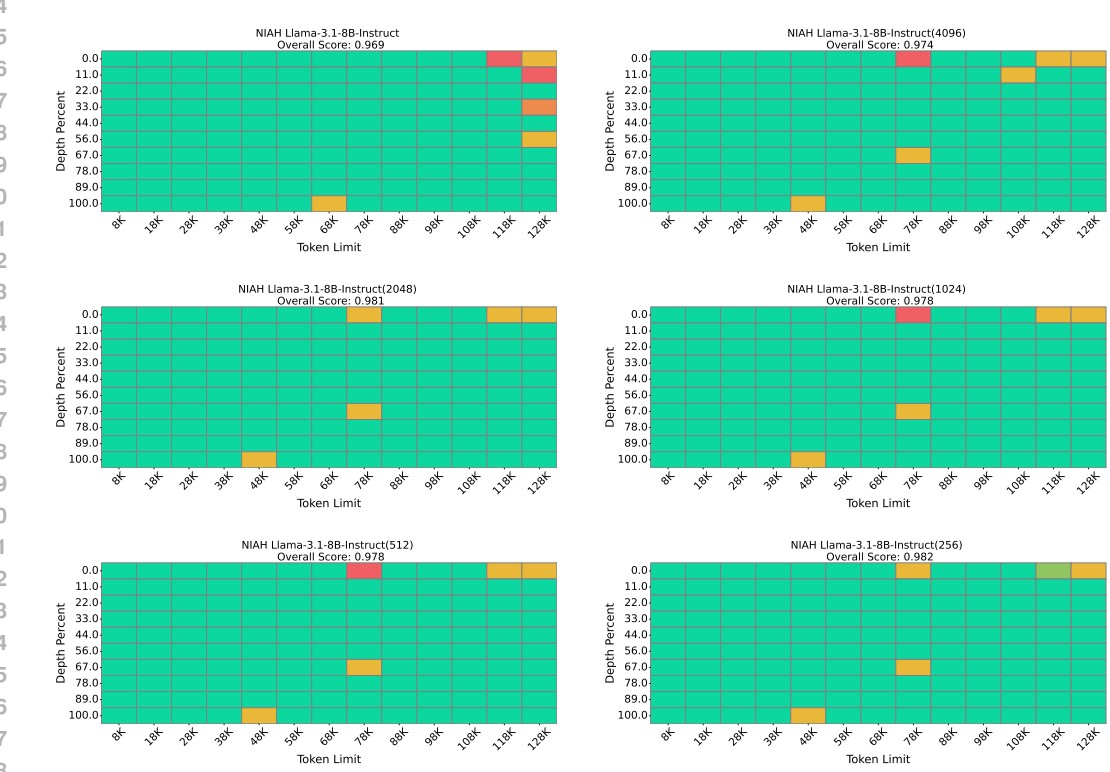

Figure 18: Compare all KV budget in SnapKV with full cache in Llama-3.1-8B-Instruct

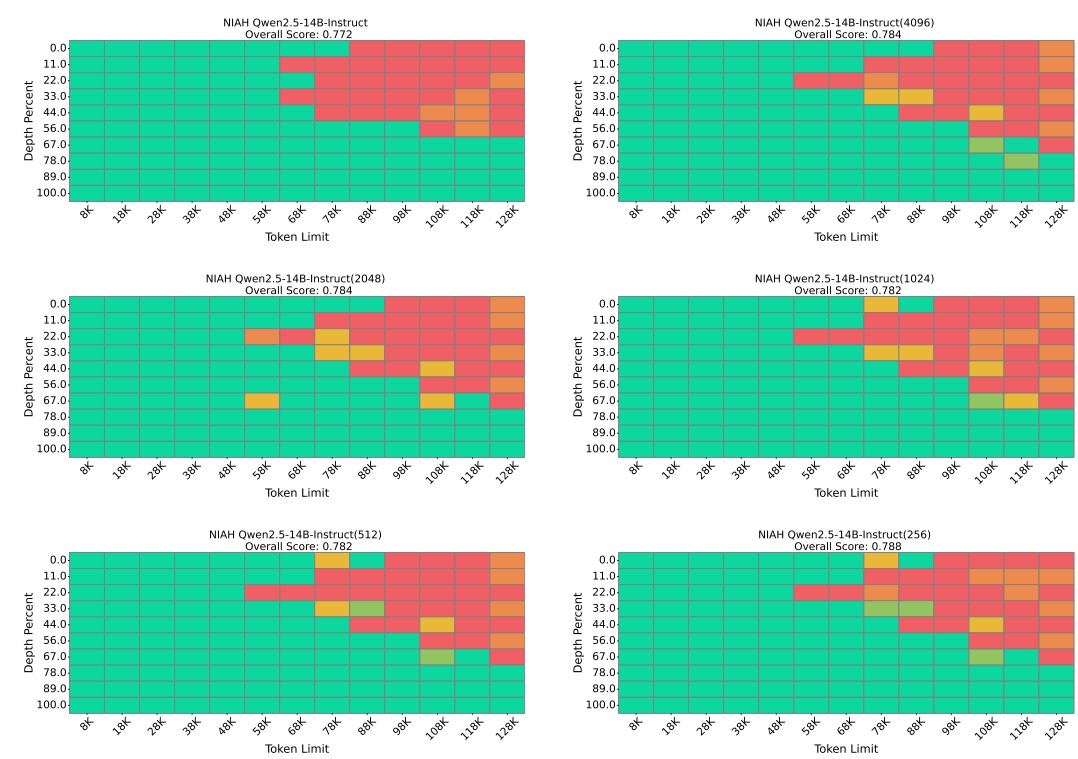

Figure 19: Compare all KV budget in SnapKV with full cache in Qwen2.5-14B-Instruct

# F    DETAIL OF ABLATION

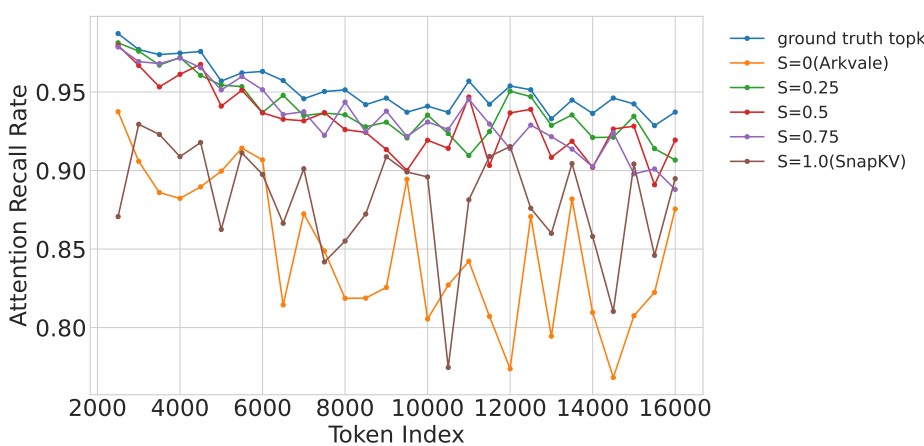

Figure 20: Attention Recall Rate of different static set ratio on 16K generation with 1K budget

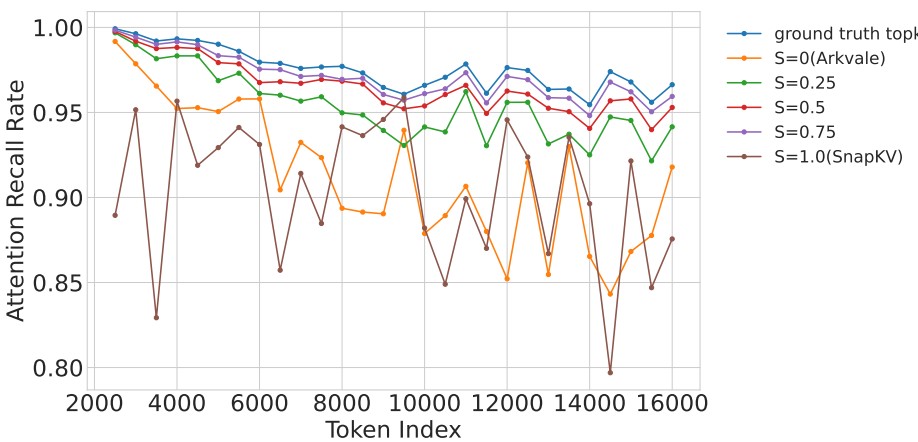

Figure 21: Attention Recall Rate of different static set ratio when KV budget is with 2K budget

## G  PRINCIPLED GUIDANCE FOR CHOOSING S

Reuse of previous topk tokens provides a fine-grained selection mechanism; however, simply reusing the topk tokens (together with local tokens) does not necessarily guarantee a high recall rate, because adjacent tokens may exhibit substantial variation in their topk attention sets. Therefore, the choice of $\mathcal{S}$ should strike a balance that maximizes the likelihood of overlap between important tokens. Based on this intuition, we design an online algorithm for determining $\mathcal{S}$.

In the prefill phase, after computing the observation-window attention scores $S$ (corresponding to line 7 of Algorithm 1), Let $S^{(1)}$ and $S^{(T)}$ denote the attention score vectors of the first and last query tokens in the prefilling phase observe window, respectively. For each head h, we compute the overlap of their topk key indices using

$$C_h = \left| \text{topk}\left( S_h^{(1)}, k \right) \cap \text{topk}\left( S_h^{(T)}, k \right) \right| \tag{4}$$

The static ratio is then obtained by averaging the overlap across all heads, $P$ denotes the page size.Then the static ratio is used in the subsequent selection process.

$$\mathcal{S} = \frac{P \cdot \left\lfloor \frac{\mathbb{E}_h[C_h]}{P} \right\rfloor}{k} \tag{5}$$

Based on this principle, we evaluate the proposed strategy $\mathcal{S}$ using Qwen2.5-14B-Instruct and Llama3.1-8B-Instruct on the LongBench dataset across various KV budgets and input length. Results in Table 4 indicate that accuracy consistently improves with larger KV budgets ($k$) but degrades as input length scales from 32K to 128K. Notably, Llama3.1-8B-Instruct outperforms Qwen2.5-14B-Instruct across all configurations, demonstrating superior robustness in sparse retrieval tasks despite having fewer parameters.

Table 4: Setting of S across input lengths and KV budgets

| KV Budget | Qwen2.5-14B-Instruct | | | Llama3.1-8B-Instruct | | |
|---|---|---|---|---|---|---|
| | 32K | 64K | 128K | 32K | 64K | 128K |
| k=512 | 0.25 | 0.25 | 0.25 | 0.625 | 0.5625 | 0.5 |
| k=1024 | 0.4062 | 0.375 | 0.3438 | 0.6875 | 0.625 | 0.5625 |
| k=2048 | 0.5 | 0.4844 | 0.4219 | 0.7344 | 0.7031 | 0.625 |
| k=4096 | 0.6016 | 0.5781 | 0.5 | 0.7891 | 0.75 | 0.6875 |

## H  THE MEASURED OVERHEAD OF RECOMPUTES THE STATIC SET AND RESTRUCTURES PAGES

After recomputing the static set and restructuring the pages, two operations are required. First, the page representations must be updated because the page organization has changed. Second, an asynchronous rewrite is issued to synchronize the restructured KV cache back to the CPU-side KV cache pool, ensuring data consistency.

Furthermore, we measured the overhead of recomputing the static set and restructuring pages (abbreviated as 2R). Figure 23 reports normal layer latency and layer latency with 2R, across different batch sizes, evaluated on both Llama3.1-8B-Instruct and Qwen2.5-14B-Instruct, and measured in both the prefill and decode phases.

The $\Delta$ shown in the figure denotes the overhead incurred by recomputing the static set and restructuring pages. In the prefill stage, this overhead is approximately 40 ms ($8\%$), imposing a negligible impact on the Time to First Token (TTFT). In the decoding stage, although the overhead is around 0.7 ms ($50\%$), our periodic update strategy allows this cost to be amortized over the interval $T$. For example, the $T = 128$ results in a $0.3\%$ overhead in each decode step.

Furthermore, the low latency achieved by our system optimizations effectively masks this overhead. Consequently, as shown in the left panel of Figure 7, we still effectively reduce the end-to-end latency.

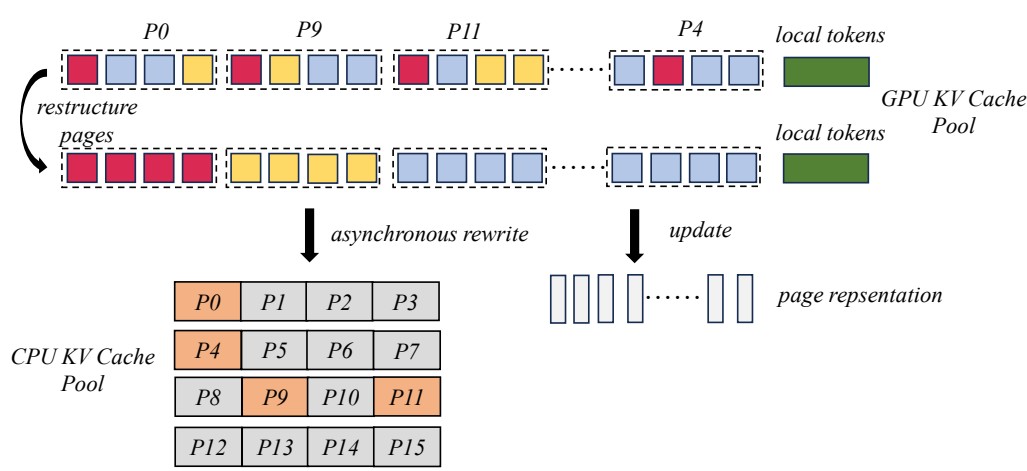

Figure 22: Detail design of recomputes the static set and restructures pages

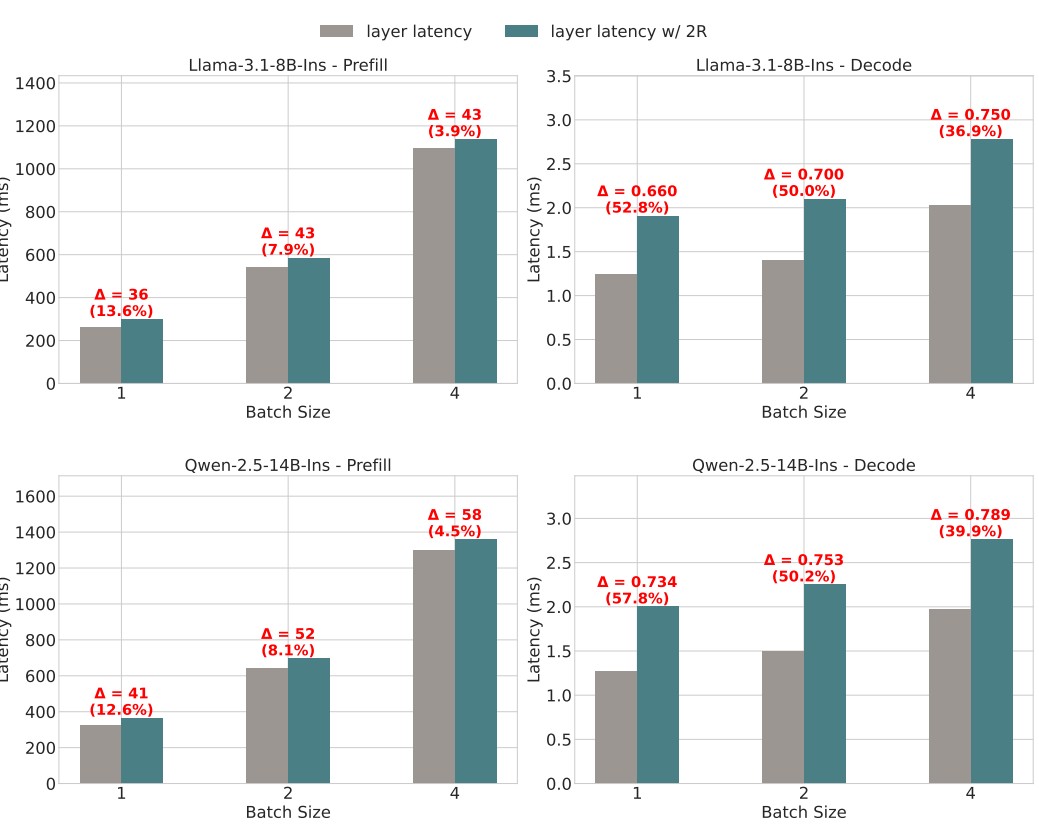

Figure 23: 2R overhead

# I  NHD AND HND KV LAYOUTS GPU EFFICIENCY

Table 5: Latency comparison of NHD vs. HND

| KV Size | Prefill NHD (ms) | Prefill HND (ms) | Decode NHD (ms) | Decode HND (ms) |
|---|---|---|---|---|
| 4,096 | 1.0358 | 1.2567 | 0.0359 | 0.0320 |
| 8,192 | 4.9863 | 4.8339 | 0.0331 | 0.0320 |
| 16,384 | 19.1479 | 19.0742 | 0.0568 | 0.0565 |
| 32,768 | 76.4648 | 77.2803 | 0.0971 | 0.0956 |
| 65,536 | 325.6366 | 317.4488 | 0.1809 | 0.1777 |
| 131,072 | 1548.4492 | 1546.6578 | 0.3658 | 0.3981 |

# J ADDITIONAL EVALUATION ON QWEN2.5-32B-INS

In this section, we provide an additional evaluation on Qwen2.5-32B-Instruct using two A100 GPUs, which represents a larger-scale model.

Table 6: Larger size model results of LongBench v2

| | | Overall | Easy | Hard | Short | Medium | Long |
|---|---|---|---|---|---|---|---|
| *Qwen-2.5-32B-Instruct (base)* | | 39.36 | 40.62 | 38.59 | 42.22 | 38.6 | 36.11 |
| | Quest | 35.39 | 30.9 | 26.95 | 33.73 | 25.25 | 26.04 |
| | ArkVale | 37.77 | 38.54 | 37.3 | 41.11 | 37.67 | 32.41 |
| *k=4096* | PRKV | **38.77** | 40.1 | 37.94 | 42.22 | 38.14 | 34.26 |
| | Quest | 33.8 | 35.42 | 32.8 | 37.78 | 33.95 | 26.85 |
| | ArkVale | 37.38 | 38.02 | 36.98 | 41.11 | 36.74 | 32.41 |
| *k=2048* | PRKV | **37.77** | 38.54 | 37.3 | 41.11 | 37.21 | 33.33 |
| | Quest | 33.4 | 34.9 | 32.48 | 37.78 | 33.02 | 26.85 |
| | ArkVale | 36.18 | 36.98 | 35.69 | 41.11 | 34.88 | 30.56 |
| *k=1024* | PRKV | **36.78** | 36.98 | 36.66 | 41.11 | 35.81 | 31.48 |

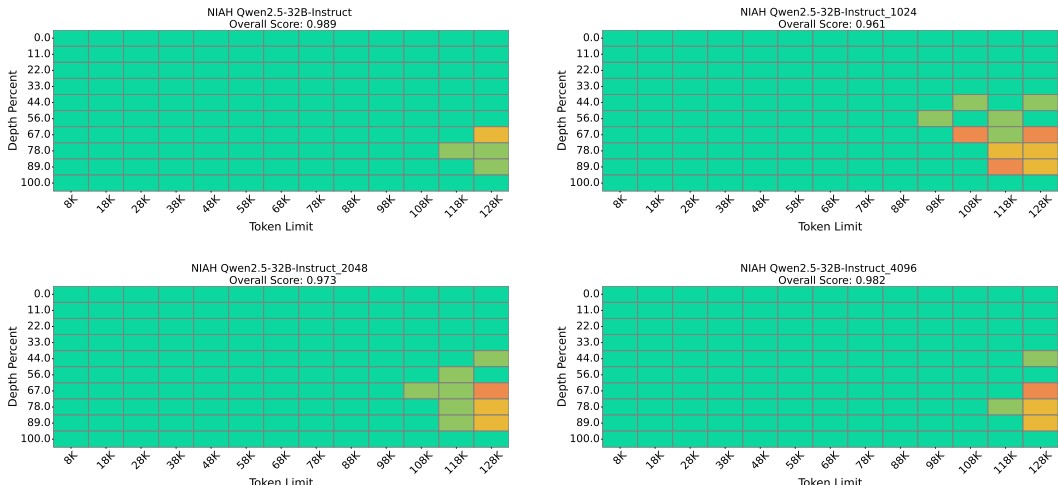

Figure 24: Compare all KV budget in PRKV with full cache in Qwen2.5-32B-Instruct

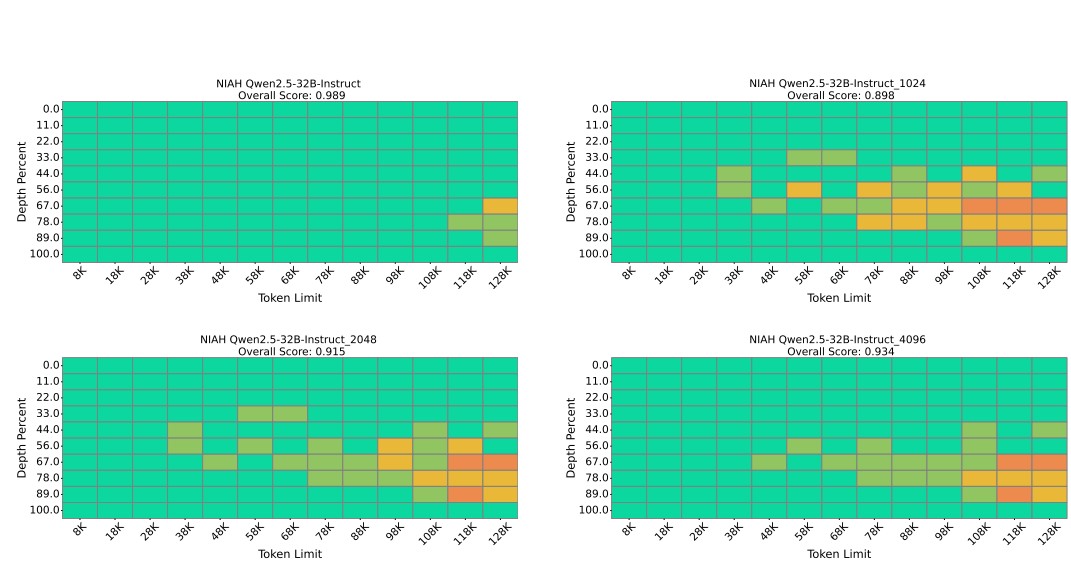

Figure 25: Compare all KV budget in Quest with full cache in Qwen2.5-32B-Instruct

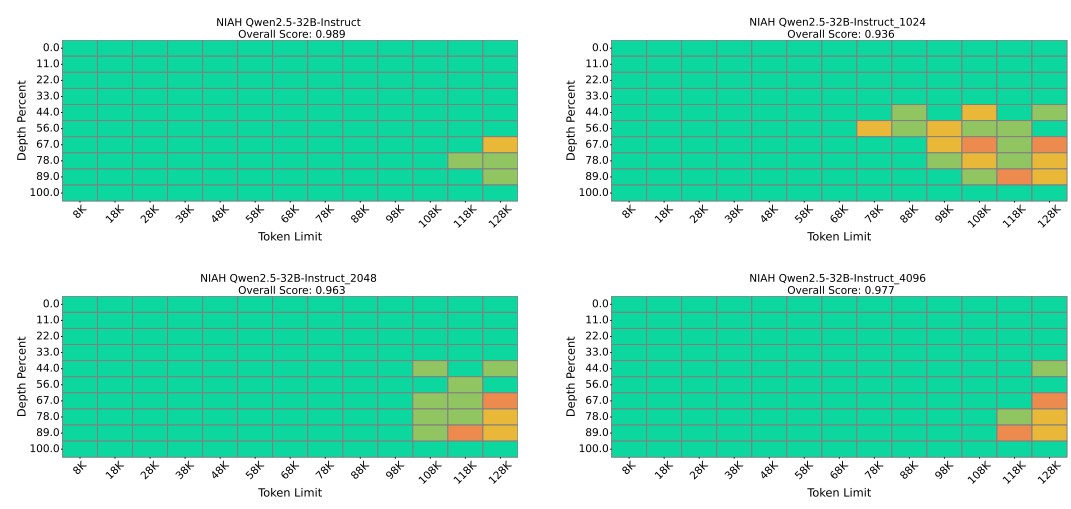

Figure 26: Compare all KV budget in Arkvale with full cache in Qwen2.5-32B-Instruct

