# OpenReview forum: "PRKV:Page Restruct KV Cache for High Accuracy and  Efficiency LLM Generation"
_ICLR.cc/2026/Conference — Submitted to ICLR 2026_

### Official Review · Reviewer_Vcit · 2025-10-31

**Soundness:** 3
**Presentation:** 3
**Contribution:** 3
**Rating:** 6
**Confidence:** 3

**Summary:**

The paper introduces PRKV, a hybrid key value retrieval framework for long context LLM inference that combines token level reuse with dynamic page level selection.
PRKV periodically reuses top k tokens from the previous step and a local window, then fills the remaining budget with page selection guided by compact page representations.
It restructures the cache so the reused tokens are stored contiguously, which simplifies indexing and reduces memory traffic.
On the systems side, PRKV uses a head major layout and a batched copy pipeline from CPU to GPU to cut transfer overheads.
Experiments on retrieval and reasoning benchmarks show accuracy close to full cache and up to 6.75x end to end speedups over prior offloading methods.

**Strengths:**

* The paper propose simple and effective hybrid design which combines prior-step token reuse with dynamic page selection to close the recall gap of page-only methods while keeping estimation overhead low.
* HND layout and batched KV transfer are broadly reusable optimizations that materially reduce TPOT and drive the reported speedups.
* Broad and convincing evaluation across multiple models and benchmarks with clear wins in both accuracy and latency.

**Weaknesses:**

* The paper relocates static tokens to the front and refreshes every T steps (Alg. 1), but the overhead (CPU time and additional PCIe traffic) of reordering KV pages/indices is not reported.
* Results stop at 14B‑parameter general models and 8B for reasoning. It remains unclear how PRKV behaves for larger size model, and models with different attention architecture (e.g. DeepSeek).
* The LongBench evaluation setup seems a bit tricky: The evaluation truncates inputs longer than the model’s context by taking half from the beginning and half from the end, then concatenating. This setup disadvantages the “full cache” baseline and can artificially make PRKV look better than “full attention” because PRKV can still select tokens from the entire document while the baseline never sees the middle. Could you elaborate on the setup?

**Questions:**

* Additionally, the score in the needle-in-a-haystack seems under-specified, making it hard to interpret results. Could you specify how you measure the success, why use Kimi API as the judge model (what about other models as the judge results)?

---

> ### Author Response · Authors · 2025-11-24
> **Response to Reviewer Vcit (Question 1/4)**
>
> **Q1: the overhead of reordering KV pages**
>
>
> After recomputing the static set and restructuring the pages, two operations are required. First, the page representations must be updated because the page organization has changed. Second, an asynchronous rewrite is issued to synchronize the restructured KV cache back to the CPU-side KV cache pool, ensuring data consistency.  The work flow in **Figure 22, page 25**. Below reports normal layer latency and layer latency with 2R, across different batch sizes, evaluated on both Llama3.1-8B-Instruct and Qwen2.5-14B-Instruct, and measured in both the prefill and decode phases.
>
> $$
> \\begin{array}{l|ccc|ccc}
> \\textbf{Model / BS} & \\textbf{Prefill(ms)} & \\textbf{Prefill w/ 2R(ms)} & \\textbf{2R Cost(ms)} & \\textbf{Decode(ms)} & \\textbf{Decode w/ 2R(ms)} & \\textbf{2R Cost(ms)} \\\\
> \\hline
> \\textbf{Llama-3.1-8B-Ins} & & & & & & \\\\
> BS=1 & 265 & 301 & 36 & 1.25 & 1.91 & 0.66 \\\\
> BS=2 & 544 & 587 & 43 & 1.40 & 2.10 & 0.70 \\\\
> BS=4 & 1103 & 1146 & 43 & 2.03 & 2.78 & 0.75 \\\\
> \\hline
> \\textbf{Qwen-2.5-14B-Ins} & & & & & & \\\\
> BS=1 & 325 & 366 & 41 & 1.27 & 2.00 & 0.73 \\\\
> BS=2 & 642 & 694 & 52 & 1.50 & 2.25 & 0.75 \\\\
> BS=4 & 1289 & 1347 & 58 & 1.98 & 2.77 & 0.79 \\\\
> \\end{array}
> $$
>
> The $2R \ Cost$ denotes the overhead incurred by recomputing the static set and restructuring pages. In the prefill stage, this overhead is approximately 40 ms ($8\%$), imposing a negligible impact on the Time to First Token (TTFT). In the decoding stage, although the overhead is around 0.7 ms ($50\%$), our periodic update strategy allows this cost to be amortized over the interval $T$. For example, the $T=128$ results in a $0.3\%$ overhead in each decode step.
>
> Furthermore, the low latency achieved by our system optimizations effectively masks this overhead. Consequently, we still effectively reduce the end-to-end latency.
>
> Detail can be found in $ \color{red}{\text{Appendix H, Page 24-25}}$

---

> ### Author Response · Authors · 2025-11-24
> **Response to Reviewer Vcit (Question 2/4)**
>
> **Q2-1: evalution for larger size model**
>
> We provide an additional evaluation on Qwen2.5-32B-Instruct using two A100 GPUs, which represents a larger-scale model.
>
> Detail can be found in $\color{red}{\text{Appendix J, Page 26-27}}$
>
>
> **Q2-2: sparse attention on models with different attention architecture(MLA)**
>
> Thank you for this insightful question; it raises a critical point for discussion.
>
> We acknowledge that DeepSeek's Multi-Head Latent Attention (MLA) architecture is highly effective. By compressing Key-Value pairs into low-rank latent matrices via a LoRA-like mechanism and projecting them back into per-head KV caches during inference, MLA significantly reduces memory footprint and enhances inference speed.
>
> However, directly applying PRKV to the MLA architecture presents significant challenges. We outline our reasoning as follows:
>
> 1. **Context of Prior Work:** Previous research on KV cache sparsity—such as Arkvale, ShadowKV, and SnapKV—has predominantly focused on Multi-Head Attention (MHA) or Grouped-Query Attention (GQA). PRKV is built upon the analysis and optimization foundations established for these standard architectures.
> 2. **Technical Constraints (Hypothesis):** To the best of our knowledge, there is currently no existing work addressing sparse methods specifically for MLA. Offering our technical perspective: standard sparse Top-K selection operates on the explicit, per-head KV cache. In contrast, MLA's compressed KV latent has head dimension with 1 (before up-projection). Applying Top-K selection directly to this latent space would imply that **all Query heads must share a single, identical sparse KV set**. We hypothesize that this lack of head-specific granularity would likely lead to a significant degradation in inference performance.
>
> Below, we give our thought about sparse attention on MLA. For all query heads, one possible idea is to group them in a manner similar to GQA: **different groups of query heads would maintain different sparse KV latents, while query heads within the same group would share a common sparse latent representation**. This design would require maintaining multiple copies of sparse KV latents.
>
> Your question is highly insightful, and **we will consider exploring sparse-attention optimization specifically for MLA in future work**.

---

> ### Author Response · Authors · 2025-11-24
> **Response to Reviewer Vcit (Question 3/4)**
>
> **Q3: Clarification of the setting when testing LongBench**
>
> We thank the reviewer for raising this concern.
> Firstly, We clarify that the truncation strategy used in LongBench—taking half of the tokens from the beginning and half from the end when the input exceeds the model’s context window—is **applied uniformly to all methods**, including the full-cache (full attention) baseline, PRKV, and all other sparse-attention baselines. All methods receive exactly the same truncated input, so the situation where “PRKV can still access the entire document while the full-attention baseline never sees the middle part” **does not occur**.
>
> Importantly, this truncation strategy is **not introduced by our implementation**. It is the **official LongBenchV2 evaluation protocol**(https://github.com/THUDM/LongBench/blob/main/pred.py), as implemented in the public repository (see `pred.py`, lines 28–30 in the official code). We strictly follow the official pipeline to ensure fairness and reproducibility.
>
> Furthermore, LongBench is designed to evaluate whether models can produce the correct answer **given the same truncated input**, which reliably reflects the behavior of sparse-attention algorithms under long-context constraints. Under this unified evaluation setup, we compare PRKV with other sparse baselines and observe that **PRKV consistently outperforms them**. This demonstrates that the performance advantage of PRKV comes from its more effective KV selection mechanism rather than any artifact introduced by the evaluation setup.
>
> We hope this clarifies the evaluation protocol and confirms that the comparison is fair and consistent with the official LongBench methodology.

---

> ### Author Response · Authors · 2025-11-24
> **Response to Reviewer Vcit (Question 4/4)**
>
> **Q4: Clarification of the Setting and Evaluation in Needle-in-a-Haystack (NIAH)**
>
> First, we clarify how we measure the NIAH success score of LLM outputs.We strictly follow the official NIAH evaluation code from https://github.com/gkamradt/LLMTest_NeedleInAHaystack. The official code predefines scoring levels of **1, 3, 5, 7, and 10**, each with its own scoring criteria, as shown in the documentation. For a given LLM output, we provide the output together with the official scoring **CRITERIA** to an evaluator LLM, and the evaluator assigns a score based on these predefined rules.
>
> ```
> CRITERIA = {
>     "accuracy": (
>         "Score 1: The answer is completely unrelated to the reference.\n"
>         "Score 3: The answer has minor relevance but does not align with the reference.\n"
>         "Score 5: The answer has moderate relevance but contains inaccuracies.\n"
>         "Score 7: The answer aligns with the reference but has minor omissions.\n"
>         "Score 10: The answer is completely accurate and aligns perfectly with the reference.\n"
>         "Only respond with a numerical score."
>     )
> }
> ```
>
> Second, while the official NIAH code uses the ChatGPT API as the judging model, our experimental environment has network limitations that prevent us from accessing it reliably. Therefore, we use the **Kimi API** as the judge model instead. In our offline tests, the judging results from the ChatGPT API and the Kimi API are **highly consistent**.

---

> ### Author Response · Authors · 2025-11-27
>
> Dear Reviewer,
>
> I hope this message finds you well. As the discussion period is nearing its end with less than three days remaining, I wanted to ensure we have addressed all your concerns satisfactorily. If there are any additional points or feedback you'd like us to consider, please let us know. Your insights are invaluable to us, and we’re eager to address any remaining issues to improve our work.
>
> Thank you for your time and effort in reviewing our paper.

---

> > ### Comment · Reviewer_Vcit · 2025-11-27
> >
> > Thanks the reviewers for the rebuttal, it has addressed most of my concerns and I decide to keep my score.

---

### Official Review · Reviewer_k1cZ · 2025-10-31

**Soundness:** 2
**Presentation:** 2
**Contribution:** 1
**Rating:** 2
**Confidence:** 3

**Summary:**

This paper introduces PRKV, a framework for page-level KV retrieval that combines static reuse and dynamic selection through a “hybrid KV selection” strategy. The authors also propose system optimizations, namely contiguous memory indexing using the HND layout and batched KV transfers to improve retrieval efficiency. PRKV reports up to 6.75× speedup and near-full-cache accuracy across multiple benchmarks.

**Strengths:**

1. Clarity and completeness:
The paper is reasonably well written and provides a comprehensive experimental evaluation across several standard long-context benchmarks.
2. Incremental algorithmic variation:
The proposed periodic static token update introduces a minor variation to existing retrieval schemes. While not particularly innovative, it reflects an effort to stabilize long-sequence performance by refreshing the static token set.

**Weaknesses:**

1. Limited algorithmic novelty:
The proposed method largely concatenates two existing paradigms of KV-cache compression, static dropping and dynamic retrieval, without introducing a fundamentally new principle. As a result, the algorithmic contribution is incremental and lacks clear conceptual advancement.
2. Questionable system design and justification:
The claimed system optimizations are not well substantiated. Although the HND layout may simplify head-wise indexing, it is unclear how this layout integrates with modern GPU attention kernels, which typically assume NHD for coalesced access and contiguous memory traversal (as used in FlashAttention and vLLM). Similarly, the batched KV transfer optimization resembles standard engineering practice rather than a substantive research contribution.

**Questions:**

1. Please justify how the HND layout efficiently supports GPU attention kernels. Most state-of-the-art implementations (e.g., FlashAttention, FlashInfer) assume NHD layout for coalesced tensor access. Has PRKV been benchmarked with real kernel implementations to confirm compatibility?
2. The proposed “static token selection” is periodically updated. Does this imply that the entire KV cache must still be retained in CPU memory to enable re-selection? If so, how does this affect the memory footprint compared to fully static methods such as SnapKV?

---

> ### Author Response · Authors · 2025-11-23
> **Response to Reviewer k1cZ (Question 1/2)**
>
> Thank you for detailed review and valuable feedback. We hope our detailed clarifications and additional experimental results will address the concerns regarding our work.
>
> **Q1: justify how the HND layout efficiently supports GPU attention kernels**
>
> Both **NHD** and **HND** layouts are supported in FlashInfer. When the KV cache is stored in **FP16**, their GPU kernel execution efficiency is nearly identical. As stated in the FlashInfer documentation (https://docs.flashinfer.ai/tutorials/kv_layout.html):
>  *“we don’t observe significant performance difference between these two layouts on fp16 kV-Cache and we prioritize `NHD` layout for better readability.”* We also conducted our own benchmarking(detailed in $\color{red}{\text{Appendix I-Table 4}}$), and the results confirm that HND does not incur additional GPU runtime overhead.
>
> | KV Size | Prefill NHD (ms) | Prefill HND (ms) | Decode NHD (ms) | Decode HND (ms) |
> | :--- | :---: | :---: | :---: | :---: |
> | 4,096 | 1.0358 | 1.2567 | 0.0359 | 0.0320 |
> | 8,192 | 4.9863 | 4.8339 | 0.0331 | 0.0320 |
> | 16,384 | 19.1479 | 19.0742 | 0.0568 | 0.0565 |
> | 32,768 | 76.4648 | 77.2803 | 0.0971 | 0.0956 |
> | 65,536 | 325.6366 | 317.4488 | 0.1809 | 0.1777 |
> | 131,072 | 1548.4492 | 1546.6578 | 0.3658 | 0.3981 |
>
> Since we use FlashInfer as attention backend，the KV-cache layout of GPU is HND. In FlashAttention and vLLM, the KV-cache layout is indeed typically **NHD**. So we can adopt a **hybrid layout**: **NHD on the GPU side and HND on the CPU side**. But our page-level fill-in mechanism remains  regardless of the GPU-side layout.
>
> Finally, our **Contiguous Memory Indexing** in PRKV explicitly optimizes the **CPU-side** KV-cache organization, which is the dominant bottleneck during sparse KV retrieval. The GPU-side layout can be flexibly chosen according to the attention backend (FlashInfer, FlashAttention, vLLM, etc.). Our current Batched KV Transfer implementation already supports **both NHD and HND** on the GPU side, ensuring **full compatibility with real kernel implementations**.

---

> ### Author Response · Authors · 2025-11-23
> **Response to Reviewer k1cZ (Question 2/2)**
>
> **Q2: Difference from static methods such as SnapKV**
>
> First, as described in our paper, PRKV stores the entire KV cache in CPU memory, while only keeping the page representations, static tokens, local tokens, and dynamically selected pages in GPU memory during decoding.
>
> Second, fully static methods such as SnapKV maintain a fixed GPU KV budget  `K`, discarding all other tokens. Thus, their GPU memory cost is always `O(K)`. PRKV follows the same rule: its GPU-side KV footprint is also `O(K)`—consisting of static tokens, local tokens, and dynamic pages. The only additional GPU component is the set of page representations, which requires only `N / page_size` vectors.  The full KV cache is retained only in CPU memory for periodic re-selection, but this incurs no additional GPU cost, and CPU memory is generally abundant and inexpensive.
>
> PRKV maintains the same GPU memory footprint as fully static methods such as SnapKV, while the additional CPU storage enables dynamic re-selection and better long-context retention, without impacting runtime memory efficiency.

---

> ### Author Response · Authors · 2025-11-26
> **Clarification of our work to Reviewer k1cZ**
>
> Regarding the reviewer’s comment on “limited algorithmic and system design novelty ”,  we respectfully disagree.
>
> First, we emphasize that the “static set” in PRKV is not dropping-based; instead, PRKV offloads the full KV cache to CPU memory, ensuring that no information is permanently discarded.
>
> **Clarification of Algorithmic Contribution**: We introduce the Attention Recall Rate as a conceptual measure to characterize the accuracy of sparse KV selection strategies. Page-level dynamic selection offers low selection overhead but inevitably performs coarse-grained KV selection, while static token-level methods provide fine-grained KV selection but incur high overhead. PRKV’s hybrid KV selection mechanism is designed to achieve the best of both worlds. Furthermore, we introduce a dynamic hybrid configuration mechanism and page restructuring, enabling PRKV to leverage the computational advantages of page-level operations while retaining fine-grained accuracy.
>
>
>
> **Clarification of System Design and Optimization**: Our system-level design and optimizations are specifically targeted at the **KV offloading** scenario, consistent with prior works such as Arkvale and ShadowKV. However, those prior works largely overlooked optimizations for **KV retrieval** under offloading—an omission that materially affects end-to-end inference latency. By carefully analyzing the retrieval pipeline, we identified two primary bottlenecks: **KV indexing** and **KV copy**. We therefore propose two targeted optimizations: **Contiguous Memory Indexing** and **Batched KV Transfer**. Contiguous Memory Indexing achieves efficient head-wise access by organizing the CPU-side KV cache in the **HND layout**; this optimization is independent of the GPU-side KV layout (see our response to Q2 for supporting evidence and explanation). Batched KV Transfer, while conceptually a common idea, requires careful, non-trivial adaptation to the sparse KV cache retrieval problem (concat fragmented KV pages, consolidating into a pinned buffer, and performing a single bulk transfer) to realize the practical latency reductions reported in our experiments, which is not just a standard engineering practice.

---

> ### Author Response · Authors · 2025-11-27
>
> Dear Reviewer,
>
> I hope this message finds you well. As the discussion period is nearing its end with less than three days remaining, I wanted to ensure we have addressed all your concerns satisfactorily. If there are any additional points or feedback you'd like us to consider, please let us know. Your insights are invaluable to us, and we’re eager to address any remaining issues to improve our work.
>
> Thank you for your time and effort in reviewing our paper.

---

> ### Comment · Reviewer_k1cZ · 2025-11-27
>
> Thanks for the clarification. I now understand your use of the HND layout for KV cache, and I will adjust my score accordingly.
>
> Regarding the algorithmic contribution, I still find the novelty somewhat limited. Besides combining static and top-K selection, the main algorithmic change appears to be introducing a periodic static token update.
>
> Moreover, in the experiments, this update interval is set to 128, which is quite a large window. This makes me question the claim that “identifying critical KV pairs adds latency to the inference process, as it resides on the CRITICAL PATH”, which is the motivation for all the system optimization. If the static set is only refreshed every 128 steps, then the reconstruction of pages does not actually need to occur at the exact decoding step—it could reasonably be deferred (e.g., compute the scores at step 128 and finalize the reconstruction at step 130, the reconstruction happens on CPU, so you can do that asynchronously). In that case, the latency added by reconstruction would not necessarily lie on the critical path.
>
> I fully acknowledge the engineering effort spent on making reconstruction fast; speed is always valuable. However, the contribution of the paper is not entirely coherent to me.

---

> > ### Author Response · Authors · 2025-11-28
> >
> > Thank you for the constructive and timely follow-up. We appreciate your careful analysis and the opportunity to further clarify the contributions of our work.
> >
> > Clarification of the algorithmic contribution:
> >
> > Although many sparse-attention methods for LLMs have been proposed, existing approaches typically suffer from one of two limitations:1. **High KV estimation overhead and low computational efficiency** (e.g., MagicPIG, InfiniGen, ShadowKV), which can even result in a higher TPOT than before sparsification. 2. **Coarse-grained KV selection**, which leads to noticeable degradation in inference accuracy (e.g., Quest)
> >
> > Our goal is to design an inference framework that achieves **really low TPOT while maintaining inference accuracy**.
> >
> > **Contribution 1:** To the best of our knowledge, we are the first to quantitatively analyze the impact of static token reuse and page-level Top-K selection on the Attention Recall Rate metric(figure 2, page 4). This analysis also reveals the feasibility of a hybrid method that jointly maintains recall and efficiency.
> >
> > **Contribution 2:** We design an online algorithm for determining $\mathcal{S}$ with low overhead. In the prefill phase, after computing the observation-window attention scores, $S$ (corresponding to line 7 of Algorithm 1), Let $ S^{(1)} $  and $S^{(T)}$ denote the attention score vectors of the first and last query tokens in the prefilling phase observe window, respectively. For each head h, we compute the overlap of their topk key indices using
> >
> > $$
> > C_{h}= \left|  \text{topk}\left(S^{(1)}_{h}, k\right) \ \cap\  \text{topk}\left(S^{(T)} _ h, k \right) \right|
> > $$
> >
> >
> > The static ratio is then obtained by averaging the overlap across all heads, $P$ denotes the page size.Then the static ratio is used in the subsequent selection process.
> > $$
> > \mathcal{S}=\frac{P \cdot \left\lfloor  \frac{\mathbb{E} _ {h} \left[C_{h}\right]} {P}  \right\rfloor}{k}
> > $$
> >
> > **Contribution 3:** Furthermore, we introduce a Static Token Selection Algorithm (page 6), and for the periodic update of static tokens, we provide a detailed design for recomputing the static set and restructuring pages (Figure 22, page 25).
> >
> >
> > The statement on page 6 — “identifying critical KV pairs adds latency to the inference process, as it resides on the CRITICAL PATH” — refers  to the fact that **performing critical-KV identification and selection at every decoding step** would lie on the critical path. It does not refer to page reconstruction.
> >
> > Your idea about performing asynchronous reconstruction on the CPU is insightful and valuable. However, this asynchronous approach faces two potential challenges:
> >
> > - the computational performance gap between CPU and GPU-performing a GEMM between the query $Q$ and long-context keys $K$ on the CPU is extremely time-consuming.
> > - the need to maintain KV-cache **data consistency** between the GPU-side cache and the CPU-side cache pool.
> > - after performing asynchronous reconstruction on the CPU, the updates to the GPU-side KV cache must proceed **sequentially across layers**, because the decoding of each layer depends on the output of the previous one.
> >
> > So, as shown in Figure 22 in page 25, we only do page restruction for in GPU KV cache and do asynchronous rewrite to synchronize the restructured KV cache back to the CPU-side KV cache pool, ensuring data consistency.

---

### Official Review · Reviewer_g8ut · 2025-11-01

**Soundness:** 2
**Presentation:** 2
**Contribution:** 2
**Rating:** 6
**Confidence:** 4

**Summary:**

The paper proposes PRKV, a hybrid sparse-attention framework for long-context LLM decoding with KV offloading. It combines a static token-level set built from an observation window (reused and periodically updated) and a dynamic page-level selection. And it reorganizes the KV layout and uses batched transfers to reduce PCIe traffic and kernel launch overhead. Experimental results report state-of-the-art quality under fixed KV budgets and impressive end-to-end speedups over prior offloading baselines.

**Strengths:**

- Addresses a practical bottleneck in long-context serving: reducing KV retrieval latency and GPU memory pressure during decoding with CPU offloading, while maintaining quality.  ￼
- Well-motivated hybrid design by reusing prior top-k with local window to boost attention recall
- System optimizations are concrete and codesigned with the proposed algorithm

**Weaknesses:**

- Some method knobs are central but their sensitivity and generalization across models or tasks are not fully explored   ￼  ￼
- A few writing issues slightly obscure otherwise solid ideas.  ￼

**Questions:**

Thank you for the submission. I like the paper overall, the hybrid selection insight is compelling, and the system side is thoughtfully engineered.  However, several descriptions are still vague or underspecified. Clarifications that would strengthen the paper:
- The authors show higher recall with a hybrid split and reuse and local window. Could you provide principled guidance for choosing S and the local window size across different models/tasks? Extra sensitivity curves beyond the current S-only ablation will be appreciated to better estimate the impact of the proposed method.  ￼  ￼
- The algorithm periodically recomputes the static set and restructures pages. What is the measured overhead of this reorganization?   ￼
- In some scenarios, PRKV matches or even beats full attention. What is the reason for this?  ￼
- The HND layout speeds up head-wise indexing, and the batched-transfer path reduces kernel launches. Is there any restriction to apply this layout? ￼

**Details Of Ethics Concerns:**

Thank you for the submission. I like the paper overall, the hybrid selection insight is compelling, and the system side is thoughtfully engineered.  However, several descriptions are still vague or underspecified. Clarifications that would strengthen the paper:
- The authors show higher recall with a hybrid split and reuse and local window. Could you provide principled guidance for choosing S and the local window size across different models/tasks? Extra sensitivity curves beyond the current S-only ablation will be appreciated to better estimate the impact of the proposed method.  ￼  ￼
- The algorithm periodically recomputes the static set and restructures pages. What is the measured overhead of this reorganization?   ￼
- In some scenarios, PRKV matches or even beats full attention. What is the reason for this?  ￼
- The HND layout speeds up head-wise indexing, and the batched-transfer path reduces kernel launches. Is there any restriction to apply this layout? ￼

---

> ### Author Response · Authors · 2025-11-23
> **Response to Reviewer g8ut (Question 1/4)**
>
> Thank you for the supportive comments and recognizing the novelty of our method. We hope our detailed clarifications  will address the concerns regarding our work.
>
> **Q1-1: Provide principled guidance for choosing S across different models/tasks**
>
> Reuse of previous topk tokens provides a fine-grained selection mechanism;however, simply reusing the topk tokens (together with local tokens) does not necessarily guarantee a high recall rate, because adjacent tokens may exhibit substantial variation in their topk attention sets. Therefore, the choice of $\mathcal{S}$ should strike a balance that maximizes the likelihood of overlap between important tokens. Based on this intuition, we design an online algorithm for determining $\mathcal{S}$.
>
> In the prefill phase, after computing the observation-window attention scores, $S$ (corresponding to line 7 of Algorithm 1), Let $ S^{(1)} $  and $S^{(T)}$ denote the attention score vectors of the first and last query tokens in the prefilling phase observe window, respectively. For each head h, we compute the overlap of their topk key indices using
>
> $$
> C_{h}= \left|  \text{topk}\left(S^{(1)}_{h}, k\right) \ \cap\  \text{topk}\left(S^{(T)} _ h, k \right) \right|
> $$
>
>
> The static ratio is then obtained by averaging the overlap across all heads, $P$ denotes the page size.Then the static ratio is used in the subsequent selection process.
>
> $$
> \mathcal{S}=\frac{P \cdot \left\lfloor  \frac{\mathbb{E} _ {h} \left[C_{h}\right]} {P}  \right\rfloor}{k}
> $$
>
>
> Based on this principle, we evaluate the proposed strategy $\mathcal{S}$ using Qwen2.5-14B-Instruct and Llama3.1-8B-Instruct on the LongBench dataset across various KV budgets and input length.
>
> | KV Budget | Qwen (32K) | Qwen (64K) | Qwen (128K) | Llama (32K) | Llama (64K) | Llama (128K) |
> | :--- | :---: | :---: | :---: | :---: | :---: | :---: |
> | *k=512* | 0.25 | 0.25 | 0.25 | 0.625 | 0.5625 | 0.5 |
> | *k=1024* | 0.4062 | 0.375 | 0.3438 | 0.6875 | 0.625 | 0.5625 |
> | *k=2048* | 0.5 | 0.4844 | 0.4219 | 0.7344 | 0.7031 | 0.625 |
> | *k=4096* | 0.6016 | 0.5781 | 0.5 | 0.7891 | 0.75 | 0.6875 |
>
> Detail can be found in **Appendix G, Page 24**
>
> **Principled guidance for choosing the local window size.**
>
> The motivation for retaining a local attention window is that nearby tokens consistently exhibit high attention weights, and therefore should always participate in the attention computation. The local window size is set as an integer multiple
> of the page size, and its optimal value may vary depending on the task. In our experiments, we follow the same local window configuration as used in Arkvale, ensuring a consistent and comparable setup.

---

> ### Author Response · Authors · 2025-11-23
> **Response to Reviewer g8ut (Question 2/4)**
>
> **Q2: The measured overhead of recomputes the static set and restructures pages**
>
> After recomputing the static set and restructuring the pages, two operations are required. First, the page representations must be updated because the page organization has changed. Second, an asynchronous rewrite is issued to synchronize the restructured KV cache back to the CPU-side KV cache pool, ensuring data consistency.  The work flow in **Figure 22, page 25**. Below reports normal layer latency and layer latency with 2R, across different batch sizes, evaluated on both Llama3.1-8B-Instruct and Qwen2.5-14B-Instruct, and measured in both the prefill and decode phases.
>
> $$
> \\begin{array}{l|ccc|ccc}
> \\textbf{Model / BS} & \\textbf{Prefill(ms)} & \\textbf{Prefill w/ 2R(ms)} & \\textbf{2R Cost(ms)} & \\textbf{Decode(ms)} & \\textbf{Decode w/ 2R(ms)} & \\textbf{2R Cost(ms)} \\\\
> \\hline
> \\textbf{Llama-3.1-8B-Ins} & & & & & & \\\\
> BS=1 & 265 & 301 & 36 & 1.25 & 1.91 & 0.66 \\\\
> BS=2 & 544 & 587 & 43 & 1.40 & 2.10 & 0.70 \\\\
> BS=4 & 1103 & 1146 & 43 & 2.03 & 2.78 & 0.75 \\\\
> \\hline
> \\textbf{Qwen-2.5-14B-Ins} & & & & & & \\\\
> BS=1 & 325 & 366 & 41 & 1.27 & 2.00 & 0.73 \\\\
> BS=2 & 642 & 694 & 52 & 1.50 & 2.25 & 0.75 \\\\
> BS=4 & 1289 & 1347 & 58 & 1.98 & 2.77 & 0.79 \\\\
> \\end{array}
> $$
>
> The $2R \ Cost$ denotes the overhead incurred by recomputing the static set and restructuring pages. In the prefill stage, this overhead is approximately 40 ms ($8\%$), imposing a negligible impact on the Time to First Token (TTFT). In the decoding stage, although the overhead is around 0.7 ms ($50\%$), our periodic update strategy allows this cost to be amortized over the interval $T$. For example, the $T=128$ results in a $0.3\%$ overhead in each decode step.
>
> Furthermore, the low latency achieved by our system optimizations effectively masks this overhead. Consequently, we still effectively reduce the end-to-end latency.
>
> Detail can be found in $ \color{red}{\text{Appendix H, Page 24-25}}$

---

> ### Author Response · Authors · 2025-11-23
> **Response to Reviewer g8ut (Question 3/4)**
>
> **Q3: In some scenarios, PRKV matches or even beats full attention.  What is the reason for this?**
>
> That is an interesting question. We believe the phenomenon is related to the small model size of  Qwen2.5 model. In Figure 6(b) and Figure 6(d), which report results of Qwen2.5-14B-Instruct on LongBench V2 and NIAH, some methods—such as PRKV and ShadowKV—even outperform the full-cache baseline. We hypothesize that this behavior arises from **overlapping or redundant information in the KV cache**, which allows selective sparse attention to occasionally filter out irrelevant or noisy tokens, leading to slightly better task performance. Similar observations have also been reported in other works when using Qwen2.5-7B-Instruct or Qwen2.5-14B-Instruct as test model in ChunkKV[1], FreeKV[2]  and Less Is More[3].
>
> [1] ChunkKV: Semantic-Preserving KV Cache Compression for Efficient Long-Context LLM Inference]
>
> [2] FreeKV: Boosting KV Cache Retrieval for Efficient LLM Inference
>
> [3] Less Is More: Training-Free Sparse Attention with Global Locality for Efficient Reasoning

---

> ### Author Response · Authors · 2025-11-23
> **Response to Reviewer g8ut (Question 4/4)**
>
> **Q4: Is there any restriction to apply this HND layout?**
>
> To summarize, there is no restriction on using the HND layout in CPU memory for storing the KV cache. Our Contiguous Memory Indexing leverages the HND layout on the CPU side to optimize sparse KV retrieval, and this design can be integrated into existing inference frameworks with minimal modification. On the GPU side, the page-level fill-in mechanism in our framework is independent of the attention backend(FlashInfer, FlashAttention, etc.) and supports both NHD and HND layouts.

---

> ### Author Response · Authors · 2025-11-27
>
> Dear Reviewer,
>
> I hope this message finds you well. As the discussion period is nearing its end with less than three days remaining, I wanted to ensure we have addressed all your concerns satisfactorily. If there are any additional points or feedback you'd like us to consider, please let us know. Your insights are invaluable to us, and we’re eager to address any remaining issues to improve our work.
>
> Thank you for your time and effort in reviewing our paper.

---

### Author Response · Authors · 2025-11-30
**Summary of  Discussion for the Area Chair**

We sincerely thank all reviewers for their careful reviews. I write this summary comment to provide the AC with the necessary information regarding the discussion phase.

I have responded thoroughly to each reviewer’s questions. Their follow-up responses and updated scores are summarized below:

- **Reviewer 1 (g8ut):** did not provide a timely reply before.
- **Reviewer 2 (k1cZ):** replied that "I now understands the use of the HND layout for the KV cache, and I will adjust my score accordingly " and he adjusted his score. He also raised additional questions regarding whether the static set refresh lies on the *critical path* of decoding and the possibility of asynchronous static token selection. We provided timely and detailed responses to address  these concerns.
- **Reviewer 3 (Vcit):** replied that “the rebuttal has addressed most of my concerns and decided to keep his score.”

**What we added during the discussion**

- Provide principled guidance for choosing S across different models/tasks, Detail can be found in Appendix G, Page 24.

- Provide measured overhead of recomputes the static set and restructures pages. Detail can be found in Appendix H, Page 24-25
- Evalution for larger size model: we provide an additional evaluation on Qwen2.5-32B-Instruct using two A100 GPUs.Detail can be found in Appendix J, Page 26-27
- Provide how the HND and NHD layout efficiently supports GPU attention kernel. .Detail can be found in Appendix I

**Clarifications**

- Algorithmic Contribution: We are the first to quantitatively analyze the impact of static token reuse and page-level Top-K selection on the Attention Recall Rate. And  we design an online algorithm for determining $\mathcal{S}$ with low overhead.Furthermore, we introduce a Static Token Selection Algorithm, and for the periodic update of static tokens, we provide a detailed design for recomputing the static set and restructuring pages.
- Apply HND KV layout in CPU side:   there is no restriction on using the HND layout in CPU memory for storing the KV cache. Our Contiguous Memory Indexing leverages the HND layout on the CPU side to optimize sparse KV retrieval, and this design can be integrated into existing inference frameworks with minimal modification. On the GPU side, the page-level fill-in mechanism in our framework is independent of the attention backend(FlashInfer, FlashAttention, etc.) and supports both NHD and HND layouts.

PRKV introduces a unified algorithm–system framework for efficient page-level KV retrieval under KV offloading. Algorithmically, it leverages a hybrid KV selection strategy that combines static KV reuse and dynamic KV retrieval to maintain high attention recall with low overhead. System-wise, PRKV  incorporates asynchronous KV reconstruction, contiguous memory indexing and batched KV transfer to  reduce latency.

We respectfully ask the Area Chair to take the reviewers’ score updates submitted during the discussion into account and to evaluate our work and responses carefully.

---

### Meta-Review · Area_Chair_1D4A · 2026-01-07

**Summary:**

The paper proposes PRKV, a framework optimizing KV cache for long-context LLMs via a hybrid selection strategy (static tokens + dynamic pages) and system-level optimizations (CPU-side HND layout, batched transfers). Reviewers g8ut and Vcit were positive, citing effective system design and solid speedups. Reviewer k1cZ initially rejected (score 2) due to misconceptions about the layout and novelty concerns. Following the rebuttal, k1cZ acknowledged the layout clarification and agreed to raise their score, though they maintained that the algorithmic contribution is incremental.

**Reviewer Concerns:**

Addressed:
- System Overhead: Authors provided data showing the periodic reconstruction overhead is negligible (approx. 0.7ms).
- HND Layout: Clarified that the HND layout is used for CPU-side indexing to speed up retrieval, while the GPU side remains compatible with standard kernels (e.g., FlashInfer).
- Experimental Setup: Clarified the LongBench truncation protocol and provided additional results for larger models (Qwen2.5-32B).
Hyperparameters: Provided a heuristic for selecting the static ratio S.

Outstanding:
- Algorithmic Novelty (k1cZ): The reviewer views the periodic static token update as an incremental variation of existing methods.
- Critical Path Necessity (k1cZ): The reviewer questioned if the rigorous optimization is necessary if the static set is only refreshed every 128 steps (suggesting it could be done asynchronously off the critical path).

**Reviewer Scores:**

Reviewer g8ut: 6 -> 6
Reviewer Vcit: 6 -> 6
Reviewer k1cZ: 2 -> 4

---

### Decision · Program_Chairs · 2026-01-26

Reject